# Enhancing Consistency of Flow-Based Image Editing through Kalman Control

**Haozhe Chi** [1]  **Zhicheng Sun** [1]  **Yang Jin** [1]  **Yi Ma** [2]  **Jing Wang** [2]  **Yadong Mu** [1*]
[1]Peking University, [2]Central Media Technology Institute, Huawei

## Abstract

Flow-based generative models have gained popularity for image generation and editing. For instruction-based image editing, it is critical to ensure that modifications are confined to the targeted regions. Yet existing methods often fail to maintain consistency in non-targeted regions between the original / edited images. Our primary contribution is to identify the cause of this limitation as the error accumulation across individual editing steps and to address it by incorporating the historical editing trajectory. Specifically, we formulate image editing as a control problem and leverage the Kalman filter to integrate the historical editing trajectory. Our proposed algorithm, dubbed Kalman-Edit, reuses early-stage details from the historical trajectory to enhance the structural consistency of the editing results. To speed up editing, we introduce a shortcut technique based on approximate vector field velocity estimation. Extensive experiments on several datasets demonstrate its superior performance compared to previous state-of-the-art methods.

## 1 Introduction

Diffusion models [44, 47, 17] have revolutionized the field of image and video generation, bringing unprecedented advancements. The recent development of the Diffusion Transformer [39] has enabled current diffusion-based models to scale up their parameters, achieving a higher level of generative capability. Notable works such as Stable Diffusion 3 [10] and Sora [5] demonstrate the remarkable potential of diffusion models in generating complex and intricate scenarios in both images and videos. Furthermore, researchers have explored the potential of diffusion models in editing tasks. For instance, DDIM inversion [45] progressively matches the target image distribution back to the original latent space, allowing for the generation of edited images by incorporating new prompt conditions. Additionally, some studies have modified attention maps [6, 18, 14] during the generation process to directly alter specific characteristics of objects. More recently, efforts have been made to integrate pretrained modules for more precise editing. For example, existing work [29] utilizes SAM [25] to extract specific regions requiring modification. Nevertheless, many diffusion-based editing methods still face challenges related to imprecise editing, primarily due to the non-linear nature of the generation trajectory.

Rectified flow [30, 31, 2], a special class of diffusion models, transforms random noise into the target distribution by linear interpolation between the two distributions. These models achieve distribution matching by constructing a velocity field, resulting in an efficient trajectory. Recent advancements have extended rectified flow models to image editing tasks. However, existing methods often struggle to balance structural consistency and editing quality effectively. For instance, Wang *et al.* [53] employ a new sampler to achieve more precise inversion and freeze specific attention values to preserve the overall semantics of edited images. While this approach maintains structural consistency, it sacrifices editing flexibility, as we will demonstrate later. Similarly, Rout *et al.* [43] utilize Linear Quadratic Regulator (LQR) control [21] to guide the editing process, and Kulikov *et al.* [27] propose using

---

*Corresponding author.

39th Conference on Neural Information Processing Systems (NeurIPS 2025).

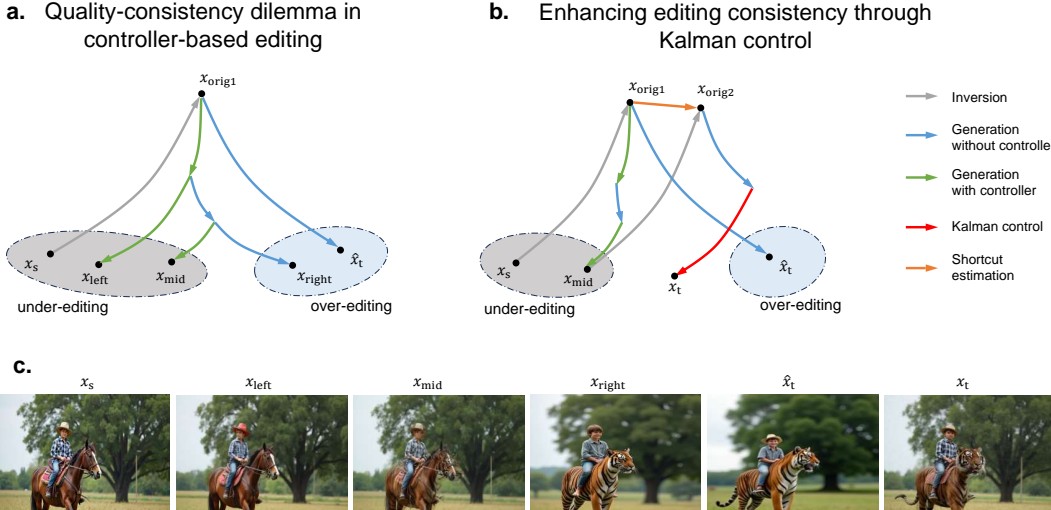

Figure 1: Conceptual illustration of a previously-developed controller-based method [43] (left) and our method (right). $x_s$ represents original image, while $\hat{x}_t$ and $x_t$ represent edited images. The controller-based method derives an optimal control strategy for different stages of the generation process. However, as illustrated in the figure for $x_{\text{left}}$, excessive control leads to failed edits, whereas $x_{\text{right}}$ demonstrates that insufficient control results in significant structural inconsistencies. Additionally, $x_{\text{mid}}$ reveals that moderate control intensity produces ambiguous image semantics. Building on these key observations, we introduce Kalman control to further suppress irrelevant semantics while maintaining structural consistency. We also propose a shortcut estimation to eliminate the need for a second inversion process. The $x_t$ figure underscores the effectiveness of our approach.

an estimated velocity transform to directly map original images to the target space. However, both methods exhibit limitations in editing quality, highlighting the need for further refinement in this area.

The primary technical contribution of this work lies in pinpointing the cause of inconsistency in image editing (*i.e.*, notable change over non-targeted image areas) as the accumulation of errors (*i.e.*, small inaccuracies can propagate forward and lead to significant deviations in the output) over individual editing steps and mitigating it by integrating the historical editing trajectory. We propose Kalman-Edit (along with its accelerated variant Kalman-Edit*), a method that addresses the Linear Quadratic Gaussian (LQG) problem [23] in optimal control theory. Our approach assumes that each velocity step in the LQG process incorporates a noise term, which can be estimated and reduced through the integration of a Kalman filter [22]. Specifically, we design an inversion algorithm that transforms the original image latent into a mixed (*i.e.*, encapsulates features from both the original and edited images). This mixed latent corresponds to a semantic blend between the source and target prompts. Subsequently, we invert the mixed latent and apply the Kalman filter to preserve structural information in the background. Our contributions are summarized as follows: (1) We introduce the Kalman filter approach to controller-based image editing. (2) Based on observations of direct LQR control, we propose a two-stage method that better unleashes the potential of Kalman control and achieves more flexible control. (3) Through experiments, our method shows high *structural consistency* and good *editing flexibility* on various editing tasks.

## 2 Related work

**Image editing with diffusion models.** Following advances of diffusion models [44, 47, 17], remarkable progress has been made in image editing via diffusion model inversion [46]. To address the time-consuming inversion computations and compounding estimation errors, various sampling and estimation strategies are developed [33, 9, 15, 35, 52, 36, 54, 20]. In addition, several work [42, 24, 28] introduce optimization methods to achieve better editing quality. Alternative editing algorithms include attention map controls [16, 50] and masking strategies [37, 8]. However, their editing quality and efficiency remain to be improved.

**Image editing with rectified flow.** Rectified flow [30, 31, 2] learns linear interpolation between source and target distributions to enable more efficient sampling of diffusion models. In addition to significant advances in rectified flow-based generation models [11, 4], its potential in image editing is gaining increasing attention [27, 38, 53, 43]. Kulikov *et al.* [27] and Patel *et al.* [38] take derivatives to produce more precise edits. Wang *et al.* [53] share attention scores in Transformer blocks to improve editing consistency. Rout *et al.* [43] use optimal controller to guide the generation trajectory. However, these efforts have not yet achieved a good balance between editing quality and consistency, as evidenced in our experiments later.

**Image editing with control theory.** Inspired by the connections between optimal control and SDEs [51, 48, 12], various sampling strategies in control theory are introduced to diffusion models for more controllable generation. Koo *et al.* [26] apply posterior sampling strategy for linear inverse problems. Rout *et al.* [43] utilize conditional sampling strategy for optimal control in the vector field. Our method advances this line of study with Kalman filter for more precise image editing.

## 3 Methodology

### 3.1 Preliminaries

**Rectified flow.** Flow-based generative models [49, 55, 30] aim to learn a probability path between the source distribution $q_0$ and the target distribution $q_1$ with a velocity field $v$. The flow starting from $x_0 \sim q_0$ to $x_1 \sim q_1$ is formulated as the following ordinary differential equation (ODE):

$$\frac{dx_t}{dt} = V(x_t, t), \tag{1}$$

where the velocity function $V$ takes the timestep $t \in [0, 1]$ and the latent variable $x_t$ as input. Rectified flow [30, 31, 2] is a special class of flow-based models defined by the linear interpolation between start point $x_0$ and endpoint $x_1$:

$$x_t = (1 - t)x_0 + tx_1. \tag{2}$$

Note that rectified flow is empirically able to generate high-resolution images with fewer sampling steps [32, 11], making it prevailing in image generation tasks.

**Editing with rectified flow.** To edit a given image $x_s$ with a source prompt $c_s$ and a target prompt $c_t$, a straightforward approach is to perform ODE inversion using Eq. (1). Let $V(x_t, t, c)$ denote the velocity function conditioned on prompt $c$, then editing follows a two-stage procedure:

$$\begin{aligned} \text{Inversion:} \quad & \frac{dx_t^s}{dt} = V(x_t^s, t, c_s), \\ \text{Generation:} \quad & \frac{dx_t^t}{dt} = V(x_t^t, t, c_t), \end{aligned} \tag{3}$$

where we first invert the image $x_s$ to structured noise $x_0$, and then perform sampling with target prompt $c_t$ for generation. However, directly applying this could lead to a significant deviation from the desired result (see $\hat{x}_t$ in Fig. 1).

**Optimal control.** To improve editing controllability, an effective approach is to formulate it as optimal control [43]. Both inversion and generation processes can be viewed as a continuous-time linear system defined over $t$ in $[0, 1]$.

$$\frac{dx_t}{dt} = Ax_t + Bu_t, \tag{4}$$

where $x_t$ represents the state of the system, $u$ serves as the controller of the system, and $A, B$ are coefficient matrices. In optimal control theory, the objective is to determine an optimal controller $u_t$ that guides the drift path to minimize the energy cost. A choice of the energy cost is a quadratic function, which corresponds to the Linear Quadratic Regulator (LQR) problem [21] as follows:

$$J_1 = x_1^\top F x_1 + \int_0^1 \left( x_t^\top Q x_t + u_t^\top R u_t \right) dt, \tag{5}$$

where $F$, $Q$ and $R$ are coefficient matrices of the system. As shown in previous work [43], solving the problem for rectified flow in Eq. (2) produces the optimal controller:

$$u_t = \frac{x_s - x_t}{1 - t},$$
(6)

where $x_s$ represents the original image for reference. Although this controller reduces the overall error during editing, it fails to preserve detail consistency for background regions, as illustrated in Fig. 1.

### 3.2 Harnessing history with Kalman filter

To improve the consistency between the edited and original images, we make a key observation that the history sequence in inversion can effectively rectify the generation trajectory. As illustrated in Fig. 1, applying control signals to latents at different steps results in distinct semantic and detail-level changes. A comparison between $x_{mid}$ and $x_{right}$ reveals that latents from later generation steps are particularly effective in recovering fine details of the original image. Additionally, we observe that early inversion latents also contribute significantly to restoring structural information. To fully leverage these latents for improved consistency, it is essential to maintain a sequence of observed latents to guide and refine the editing process. **This approach naturally aligns with Kalman control, where observations are used to refine estimations and achieve more accurate outcomes.**

Inspired by optimal control theory, we reformulate the editing process as a Linear Quadratic Gaussian (LQG) problem [23] and introduce the Kalman control method [22]. To be specific, in LQG control theory, the system's evolution at each timestep includes noise terms, which can be mitigated by refining the trajectory using Kalman filter, *i.e.*,

$$\frac{dx_t}{dt} = Ax_t + Bu_t + w_t, \quad y_t = Hx_t + \sigma_t,$$
(7)

where $y_t$ represents the measurement sequence, $A$, $B$ and $H$ are the coefficient matrices of the system, while $w_t$ and $\sigma_t$ denote the noise terms of system state estimations. With the application of the Kalman filter, the total cost function to be minimized is given by:

$$J_2 = \mathbb{E}\left[ x_1^\top F x_1 + \int_0^1 \left( x_t^\top Q x_t + u_t^\top R u_t \right) dt \right],$$
(8)

where $\mathbb{E}$ refers to the expectation of the following terms. Importantly, expectation is necessary in this context, as our goal is to mitigate the impact of noise terms when minimizing the cost function $J_2$. To accurately estimate the expectation in $J_2$, we utilize the following Kalman filter equations:

$$\begin{aligned}
K_k &= P_{k-1}H^T(HP_{k-1}H^T + T)^{-1}, \\
x_k &= Ax_{k-1} + Bu_k + K_k(y_k - Hx_{k-1}), \\
P_k &= (I - K_kH)P_{k-1}.
\end{aligned}$$
(9)

In specific, given the covariance matrix $P_{k-1}$, state $x_{k-1}$, noise $T$, controller $u_k$ and measurement $y_k$, it first computes the Kalman gain $K_k$. These values are then used recursively to obtain the updated terms $x_k$ and $P_k$. Consequently, all the filtered latents $x_k$ can be computed using these equations. Here, the noise term is estimated and corrected by multiplying the Kalman gain $K_k$ with the innovation term $(y_k - Hx_{k-1})$. Through proper derivation and analysis, we arrive at the following proposition (see Appendix A for details):

**Proposition 1.** *With proper initialization of system coefficients ($P_0$, $H$, $T$), the Kalman control process shown in Eq.* (9) *converges.*

As shown in Fig. 1, achieving both high generation quality and structural consistency presents a significant challenge in current controller-based editing methods. Addressing this issue requires establishing an effective measurement sequence. To sufficiently incorporate historical information, we carefully construct the measurement sequence $\{y_k\}_{k=1}^l$ of length $l$, integrating latents from the inversion process:

$$y_k = \text{Inv}(x_s, k), \quad 1 <= k <= l,$$
(10)

where $\text{Inv}(x, k)$ denotes the $k^{th}$ latent obtained during the inversion process described in Eq. (3), and the resulting measurement sequence $\{y_k\}_{k=1}^l$ are incorporated in the computation of the Kalman filter to control the generation process.

---

**Algorithm 1 Kalman-Edit and Kalman-Edit***

---

**Input:** original image $x_\text{s}$, total step $N$, source prompt $c_\text{s}$, target prompt $c_\text{t}$, Inversion function $\text{Inv}(\cdot, \cdot)$
    **# Stage 1: editing with optimal controller**
    $x_\text{orig1} \leftarrow \text{Inv}(x_\text{s}, N)$
    Generate $x_\text{mid}$ from $x_\text{orig1}$ with controller in Eq. (6)
    **# Stage 2: editing with Kalman filter**
    **if** Kalman-Edit **then**
        $x_\text{orig2} \leftarrow \text{Inv}(x_\text{mid}, N)$
    **else if** Kalman-Edit* **then**
        $x_\text{orig2} \leftarrow x_\text{orig1} + x_\text{mid} - x_\text{s}$                    ▷ shortcut in Section 3.4
    Compute measurement $\{y_k\}_{k=1}^{l}$ according to Eq. (11)
    Generate $x_\text{t}$ from $x_\text{orig2}$ with Kalman filter in Eq. (12)
**Output:** $x_\text{t}$

---

Following this formulation, the entire Kalman control process proceeds as follows: First, we construct the measurement sequence using Eq. (10). Next, we iteratively update the Kalman gain $K_k$ and compute the corresponding innovation term $y_k - H x_{k-1}$ to regulate the generation sequence $x_k$ using Eq. (9). This way, the generation process is conditioned on measurement from the original image before editing, thus preserving more details.

### 3.3 Proposed algorithm: Kalman-Edit

Applying Kalman control to flow-based editing presents two primary challenges: (1) As shown in Fig. 1, determining the appropriate timesteps for applying the Kalman filter is challenging. If too many timesteps undergo the filtering process, editability may be compromised, leading to edited results that fail to faithfully reflect the target prompt. (2) Artifacts and blurring often arise when directly applying Kalman control through a single inversion and forward process. This occurs because the filtering equations can inadvertently guide the trajectory toward an intermediate state between the original image distribution and the target distribution. Such a middle state often corresponds to poor image quality, resulting in undesirable visual artifacts.

**Two-stage image editing.** To address both challenges, we propose a two-stage algorithm for generating high-quality edited images using Kalman control, as outlined in Algorithm 1. In the first stage, our goal is to generate an intermediate latent that encapsulates the semantics of both the original and target prompts. Next, we generate the intermediate latent $x_\text{mid}$ by applying the controller at appropriate timesteps. In the second stage, we first invert $x_\text{mid}$ to obtain the second original latent $x_\text{orig2}$. We then apply the Kalman filter, as described in Eq. (9) to the generation process in order to filter out undesired semantic information irrelevant to the target prompt, and reintroduce history information from the original image $x_\text{s}$. This approach results in a target image $x_\text{t}$ that retains the structural information of $x_\text{s}$ while adhering to the target prompt $c_\text{t}$, as desired.

We further adapt the measurement sequence construction to the two-stage editing scheme. Since the original image $x_\text{s}$ and edited intermediate image $x_\text{mid}$ both contain desired information (*i.e.*, the original detail and target semantics), we curate the measurement sequence by collecting inversion trajectories from both images $x_\text{s}$ and $x_\text{mid}$. Specifically, we introduce a hyperparameter $\delta$ and define the measurement sequences $\{y_k\}_{k=1}^{\delta-1}$ and $\{y_k\}_{k=\delta}^{l}$ to capture structural information from the first and second inversion processes, respectively. They are computed by the following inversion processes:

$$
\begin{aligned}
y_k &= \text{Inv}(x_\text{s}, k), \ \ 1 <= k <= \delta - 1, \\
y_k &= \text{Inv}(x_\text{mid}, k), \ \ \delta <= k <= l.
\end{aligned}
\tag{11}
$$

By incorporating the two inversion sequences and integrating them into our measurement sequence, it enables the recovery of diverse structural and semantic information through Kalman control. Note that this construction scheme is also compatible with our accelerated version without two inversion passes (Section 3.4), in which case we simply set the $\delta$ value to $l + 1$.

**Kalman filter phases.** Another crucial consideration in our approach is selecting the appropriate timesteps for applying both the controller and the Kalman filter effectively. A key observation is that rectified flow models exhibit behavior similar to traditional diffusion models. As noted in prior

Table 1: Quantitative comparison on SFHQ datasets among flow-based editing models. See the main text for the definitions of the performance metrics. The highest value in each column is highlighted in bold.

| | Face Rec. ↓ | CLIP-I ↑ | LPIPS ↓ | CLIP-T ↑ | DreamSim ↓ |
|---|---|---|---|---|---|
| RF-Edit | 0.4051 | 0.8984 | 0.1562 | 0.2910 | 0.1591 |
| RF-Inversion | 0.4325 | 0.8927 | 0.1720 | **0.3012** | 0.1889 |
| FlowEdit | 0.4856 | 0.8579 | 0.1687 | 0.2905 | 0.2375 |
| FlowChef | 0.4013 | 0.8769 | 0.1401 | 0.2832 | 0.1487 |
| Kalman-Edit | **0.3958** | **0.9167** | **0.1332** | 0.2921 | **0.1408** |
| Kalman-Edit* | 0.4696 | 0.8871 | 0.1892 | 0.2936 | 0.2227 |

Table 2: Quantitative comparison on HQ datasets among flow-based editing models.

| | CLIP-T ↑ | CLIP-I ↑ | LPIPS ↓ | DINO ↑ | DreamSim ↓ |
|---|---|---|---|---|---|
| RF-Edit | 0.1842 | **0.9141** | 0.2383 | **0.8197** | 0.1492 |
| RF-Inversion | 0.1825 | 0.9033 | 0.3074 | 0.7963 | 0.1662 |
| FlowEdit | 0.1877 | 0.8813 | 0.2846 | 0.7467 | 0.2238 |
| FlowChef | 0.1928 | 0.9023 | 0.2925 | 0.8053 | 0.1537 |
| Kalman-Edit | **0.1943** | 0.9062 | **0.2345** | 0.7929 | **0.1353** |
| Kalman-Edit* | 0.1870 | 0.8696 | 0.3615 | 0.7123 | 0.2276 |

work [56], image generation in diffusion models progresses through distinct phases: the early stages primarily establish semantic content, while finer details are refined in the later stages. This pattern is also evident in rectified flow models. Applying the controller across too many timesteps during the refinement phase would overly constrain the output distribution, making it too similar to the original image and thereby limiting effective editing. To ensure high-quality editing, we apply the controller primarily during the early phase of generation when semantic information is being formed. Likewise, to maximize the effectiveness of Kalman filtering, we apply it during the refinement stage, where it can best aid in recovering structural details.

**Implementation for flow models.** In the above derivation we consider the evolution of $x_k$. However, flow-based generative models are often parameterized by velocity instead of $x$-prediction. Therefore, in practice, we apply the Kalman control updates in the following manner:

$$
\begin{aligned}
x_{t_{k+1}} &= x_{t_k} + (t_{k+1} - t_k)v'_{t_k}, \\
v'_{t_k} &= \mu v_{t_k} + \lambda u_{t_k} + (1 - \mu - \lambda)K_k(y_k - Hx_{t_k}).
\end{aligned}
\tag{12}
$$

where $v_{t_k} = V(x_{t_k}, t_k)$ is the predicted velocity, $u_{t_k}$ is the optimal controller given by Eq. (6), and $\mu$ and $\lambda$ are coefficients balancing their contributions. See Algorithm 2 for complete update details.

### 3.4 Kalman-Edit*: acceleration with shortcut

To avoid the computational cost of performing the inversion process twice, we accelerate the estimation of $x_{\text{orig2}}$ by leveraging the parallelogram law of vectors. This approach is justified in many editing scenarios, such as local area modifications, where the difference between $x_{\text{mid}}$ and $x_s$ is minimal. Under the assumption that both source and target latents are normalized and exhibit similar variance, a first-order approximation in the vector field yields a sufficiently accurate estimate of $x_{\text{orig2}}$:

$$
x_{\text{orig2}} = x_{\text{orig1}} + (x_{\text{mid}} - x_s).
\tag{13}
$$

This avoids the second inversion process and turns out to be efficient through experiments.

## 4 Experiments

### 4.1 Evaluation protocols

**Datasets.** The experimental evaluation is conducted across four widely used datasets: SFHQ [3], HQ [19], ZONE [29] and DIV2K [1]. The SFHQ dataset consists of 425,000 high-quality human facial images. The HQ dataset[2] is a synthetic editing benchmark containing approximately 200,000

---

[2]https://thefllood.github.io/HQEdit_web/

Table 3: Quantitative evaluations on ZONE and DIV2K datasets. See main text for more details about the performance metrics.

|  | CLIP-T ↑ | CLIP-I ↑ | LPIPS ↓ | DINO ↑ | DreamSim ↓ |
|---|---|---|---|---|---|
| SDEdit | 0.2754 | 0.9264 | 0.1908 | 0.8547 | 0.1148 |
| P2P | 0.2773 | 0.9209 | 0.1568 | 0.8186 | 0.1519 |
| MasaCtrl | 0.3103 | 0.9179 | 0.1580 | 0.8397 | 0.1635 |
| DDPM-Inv | 0.2847 | 0.9063 | 0.1734 | 0.8215 | 0.1742 |
| RF-Edit | 0.2964 | 0.8926 | 0.2039 | 0.7986 | 0.1776 |
| RF-Inversion | 0.2844 | 0.8919 | 0.2491 | 0.7974 | 0.1536 |
| FlowEdit | 0.3096 | 0.8687 | 0.2269 | 0.7671 | 0.2319 |
| FlowChef | 0.3025 | 0.8831 | 0.2563 | 0.7456 | 0.2471 |
| Kalman-Edit | 0.2957 | **0.9492** | **0.1407** | **0.9141** | **0.0793** |
| Kalman-Edit* | **0.3220** | 0.8986 | 0.2488 | 0.8237 | 0.1454 |

Table 4: Comparison of CLIP-I (left) and LPIPS scores(right) for different Kalman filter strengths and steps evaluated on the ZONE dataset.

| Filter strength / Added steps | 15-18 | 15-22 | 15-27 | Filter strength / Added steps | 15-18 | 15-22 | 15-27 |
|---|---|---|---|---|---|---|---|
| 0.1 | 0.8770 | 0.9219 | **0.9346** | 0.1 | 0.2433 | 0.2035 | **0.1487** |
| 0.2 | 0.9043 | 0.9282 | 0.9226 | 0.2 | 0.2325 | 0.1944 | 0.1521 |
| 0.3 | 0.9014 | 0.8921 | 0.9079 | 0.3 | 0.2284 | 0.2008 | 0.1886 |

images generated through DALL-E 3 and GPT-4V. The ZONE dataset features 100 images designed for object insertion, editing, and removal tasks. DIV2K serves as a standard benchmark for super-resolution tasks, comprising 1,000 real-world images. Due to practical computational constraints, we evaluate our approach on the subsets of these benchmarks, including 1,200 images from SFHQ, 320 images from HQ, and 105 images from ZONE and DIV2K. Following the setting of Rout *et al.* [43], we employ an instruction prompt that adds glasses to all face images in the SFHQ dataset.

**Metrics.** Six metrics are employed to evaluate both editing quality and consistency. For editing quality, CLIP-T [40] is adopted to measure the semantic adherence between edited image and input prompts. Meanwhile, we also use Face Rec. metric to quantify identity similarity on the face-specific SFHQ dataset. Regarding editing consistency, CLIP-I and DINO [7] measure high-level semantic similarity, while LPIPS [57] captures low-level similarity such as pixel-level details. Moreover, Dreamsim [13] is responsible for evaluating mid-level similarity, including image layout.

**Baselines.** Our method is compared against eight image editing baselines spanning rectified flow and diffusion models. For rectified flow-based editing, we consider RF-Edit [53], RF-Inversion [43], FlowEdit [27] and FlowChef [38]. For diffusion-based editing counterparts, we compare against SDEdit [34], P2P [16], MasaCtrl [6] and DDPM-Inv [18]. To ensure a fair comparison, we follow the original recommended hyperparameter settings (*e.g.*, where to add the controller) for all baselines.

**Implementation details.** For flow-based editing, we use FLUX.1 dev [4] with $N = 28$ sampling steps. For diffusion-based editing, we use Stable Diffusion 1.4 [41] with $N = 50$ sampling steps. The measurement length $l$ is set to 14, and $\delta$ is 6 by default. More details are provided in Appendix B.

## 4.2 Experimental results

**Quantitative results**. As demonstrated in Table 1 and Table 2, our method maintains high facial similarity after editing on the SFHQ dataset and effectively adheres to complex editing prompts on the HQ dataset. The basic version of Kalman-Edit demonstrates strong performance, achieving state-of-the-art results in most metrics. And Kalman-Edit* (the accelerated variant) produces comparable results on SFHQ and HQ datasets. This discrepancy can be attributed to the fact that Kalman-Edit* directly estimates the second original latent, which may result in the loss of low-level structural information. As shown in Table 3, Kalman-Edit and Kalman-Edit* also outperform the four baseline methods across most metrics on ZONE and DIV2K datasets, showing the effectiveness of our method.

**Ablation analysis**. In this section, we conduct ablation experiments to determine the optimal filter strength and steps at which the filter is applied, as well as to highlight the importance of Kalman control in structural preservation. Following previous research [56], we observe that the generation process of rectified flow can also be divided into two stages: semantic formation and refinement.

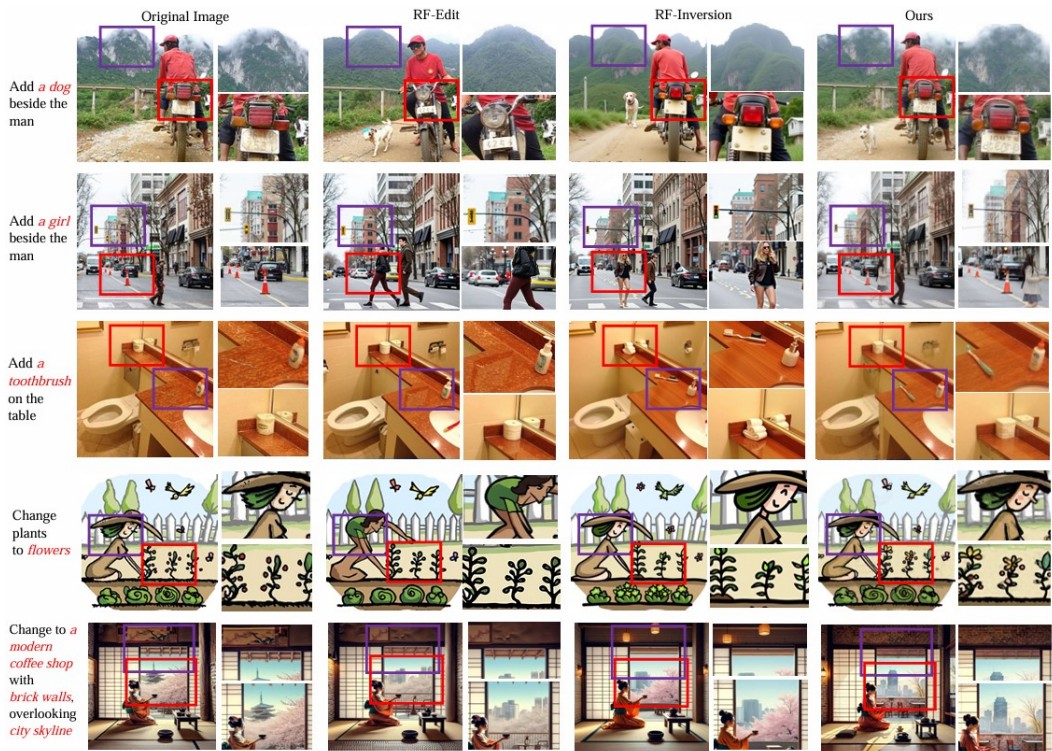

Figure 2: Comparison of structure preservation and editing quality on ZONE and DIV2K dataset. The top three rows demonstrate that our method effectively preserves local details, ensuring strong structural consistency. Meanwhile, the bottom two rows highlight its ability to adhere to the target prompt, accurately incorporating elements such as flowers and brick walls. They illustrate that our method achieves both better structure preservation and editing quality. Better viewing when enlarged.

Table 5: Ablation study of Kalman filter in LQR-based control method on ZONE dataset.

| Metrics | CLIP-T ↑ | CLIP-I ↑ | LPIPS ↓ |
|---|---|---|---|
| w/o Kalman filter | 0.2952 | 0.8784 | 0.2695 |
| w/ Kalman filter | **0.2961** | **0.9346** | **0.1487** |

Since the Kalman filter is designed to refine structural details, we apply Kalman control during the refinement stage, which corresponds to the latter half of the generation steps. As shown in Tables 4 and 5, the best CLIP-I and LPIPS scores are achieved when using a relatively small filter strength combined with a large number of added steps. Also, from each row of Table 4 we conclude that more steps of Kalman control help recover more pixel-level details. From each column of Table 4, we observe that small strength helps to generate structural details more efficiently and smoothly, while large strength causes significant performance drop. This indicates that a longer measurement sequence enhances structural details, while lower filter strength steers the trajectory toward a higher-quality distribution. Overall, our method is effective across a fairly wide range of hyperparameters.

To validate the effectiveness of our approach, we conduct ablation experiments to assess the impact of Kalman control. Specifically, we compare our method against the RF-Inversion baseline without the Kalman control. As shown in Table 5, our method achieves higher CLIP-I and LPIPS scores, indicating superior structural consistency. Furthermore, the CLIP-T metric confirms that our approach maintains high editing quality, demonstrating its advantages in both structure and semantics.

**Qualitative results**. Our qualitative comparison results are presented in Figures 2 and 3, showcasing both real-world and synthetic images to demonstrate our method's ability to maintain high structural consistency and editing quality. In particular, Fig. 2 compares the structural preservation capabilities of our method against baseline approaches. As shown, our method retains more local details than the baselines. For instance, in the second row of Fig. 2, only our method successfully recovers the

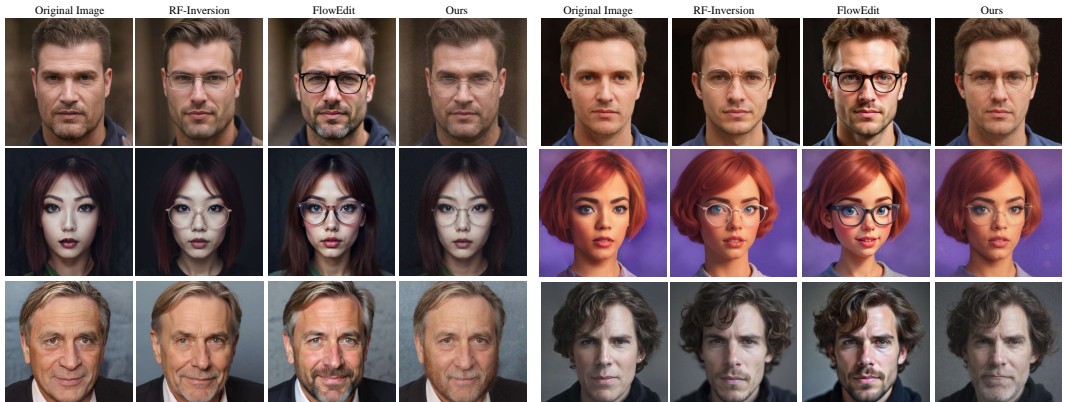

Figure 3: Qualitative results obtained using human face images from SFHQ dataset. The first two rows are edited with target prompt "A person wearing glasses" and the last row is edited with target prompt "A person with beard". The edited results are compared with baseline methods, demonstrating our approach's superior ability to preserve the structural details of human faces (*i.e.*, our method produces edited images of higher fidelity, recovering facial features more accurately than baseline approaches).

traffic cones on the road. Similarly, in the last row, our edited results accurately incorporate multiple elements specified in the target prompt, demonstrating our method's flexibility in handling longer and more complex prompts. In contrast, baseline methods struggle to recover structural details while maintaining overall editing quality. Compared to the baselines, our method significantly outperforms in preserving the structure of the original images while effectively editing the desired areas. Furthermore, to demonstrate the effectiveness of our method in preserving structural information across various image structures, we present editing results on human faces in Fig. 3. Specifically, we first modify the images by adding glasses to each face, following the approach in [43]. To further validate the generality of our method, we then edit the images by adding beards to each face. The edited results maintain a high degree of structural similarity to the original faces, highlighting our method's ability to preserve facial structural consistency. For additional qualitative results on tasks such as style transfer and scene editing, please refer to Appendix D.

**Computational efficiency.** Figure 4 compares the time usage of our approach against flow-based editing methods. Since RF-Edit takes much time to load models, we report the time cost from the starting point of inversion to the time editing is completed to ensure the fairness. From the comparison, we observe that FlowChef has the fastest editing speed. Our proposed Kalman-Edit* is slower than RF-Inversion and FlowChef, but faster than all other baselines. This indicates that our algorithm strikes a competitive balance between editing performance and efficiency.

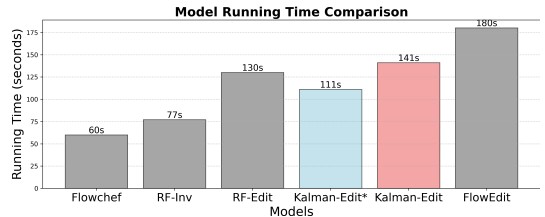

Figure 4: Running time comparison of different flow-based methods highlighting efficiency.

**Failure cases**. For tasks such as object removal, our method requires hyperparameter tuning to achieve optimal results. This necessity arises from our design that leverages historical inversion latents as the measurement sequence to rectify the final generation process. As a consequence, the original details and residual artifacts from the original image tend to appear in edited images at the default control strength settings, as illustrated by the boat example in Appendix G.

**Style transfer and Scene editing**. To further evaluate our method on a wider range of tasks, such as style transfer and scene editing, we present additional qualitative results. As illustrated in Fig. 5, our method demonstrates strong capability in transforming an old rusty room with a cement floor into a simple and elegant room with a wooden floor, converting a castle into a Disney-style cartoon scene, transforming a house into a white church with stained-glass windows, and replacing a beach background with snow-covered mountains. Moreover, our method effectively preserves structural consistency, maintaining both local details and the overall spatial layout. For example,

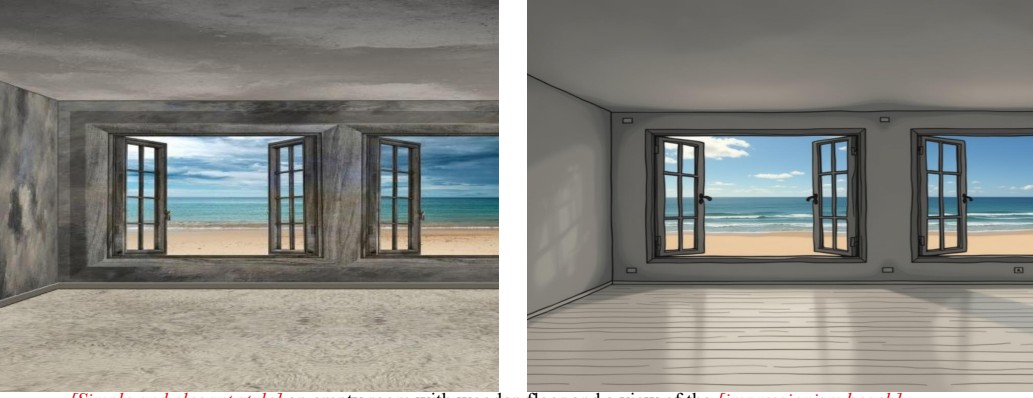

*[Simple and elegant style]* an empty room with wooden floor and a view of the *[impressionism beach]*

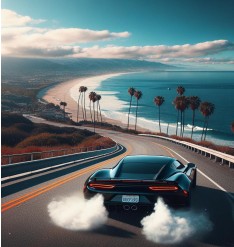 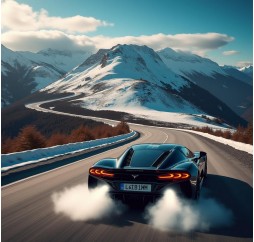 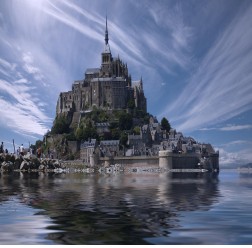 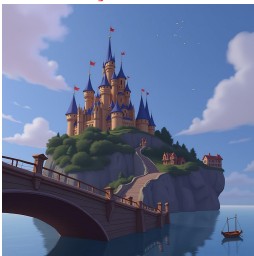

Change the *beach* in the background to a *snow mountain*     Change the castle to *Disney cartoon style* and add *a bridge*

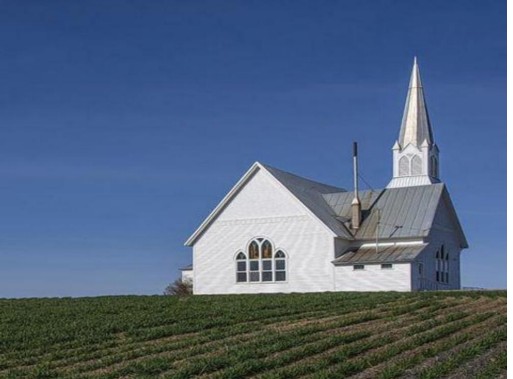 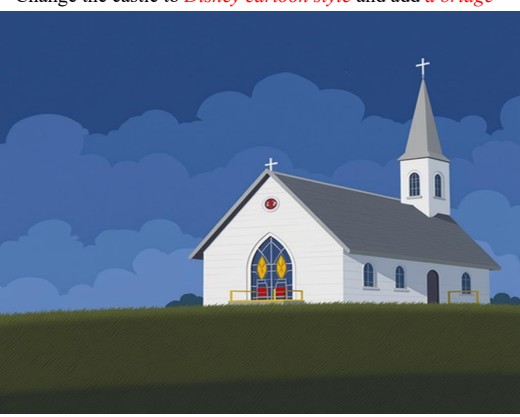

*[Stained glass window of]* a white *[cartoon]* church sits on a hill in a field

Figure 5: High-resolution qualitative results of style transfer and scene editing tasks. The left image is the original input and the right one is the edited result. As illustrated above, our method achieves precise prompt adherence and delivers high-quality editing outcomes.

the edited result in the top case of Fig. 5 retains the room's geometric structure, while in the car case, it accurately preserves the road direction and car position. These results demonstrate that our method can successfully handle a broad variety of complex tasks and produce high-quality, structure-preserving outputs. Additional discussions on broader tasks and high-resolution visualization results are provided in Appendix D and Appendix E.

## 5   Conclusion

In this paper, we propose Kalman-Edit, a training-free flow-based image editing method based on optimal control theory. Existing rectified flow editing methods struggle to balance structural consistency and editing quality. To address this challenge, we derive fundamental equations from Linear Quadratic Gaussian (LQG) control, effectively utilizing history information in the editing trajectory with a Kalman filter-based algorithm. Through extensive experiments, we demonstrate that Kalman-Edit achieves superior structural consistency while maintaining high editing quality.

**Acknowledgement:** This work is supported by a grant from Huawei (No. TC20240821013_01).

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

# A  Additional theoretical results: Proof of convergence for Kalman control iteration

We show that the error covariance sequence $P_k$ decreases at each iteration, ensuring convergence. According to the Kalman iteration in Eq. (9), the update rule is given by:

$$P_k = (I - K_k H)P_{k-1}, \tag{14}$$

$$K_k = P_{k-1}H^T S_k^{-1}, \tag{15}$$

$$S_k = HP_{k-1}H^T + T. \tag{16}$$

Substituting Eq. (15) into Eq. (14), we obtain:

$$P_k = P_{k-1} - P_{k-1}H^T S_k^{-1} HP_{k-1}. \tag{17}$$

Defining $M = P_{k-1}H^T S_k^{-1} HP_{k-1}$, we need to show $M$ is positive semidefinite to conclude $P_k \preceq P_{k-1}$.

Assuming $P_0$ is positive definite and $P_{k-1}$ remains positive definite, and also that $H$ is positive semidefinite and $T$ is positive definite with proper initialization, we first show that $S_k^{-1}$ is positive definite. According to Eq. (16), since $P_{k-1}$ is positive definite, $HP_{k-1}H^T$ is positive semidefinite. As $T$ is positive definite, their sum $S_k$ is also positive definite, implying $S_k^{-1}$ is positive definite.

To establish $M \succeq 0$, for any nonzero vector $x$, we compute

$$x^T M x = x^T P_{k-1}H^T S_k^{-1} HP_{k-1}x. \tag{18}$$

Letting $y = HP_{k-1}x$, this simplifies to

$$x^T M x = y^T S_k^{-1} y. \tag{19}$$

Since $S_k^{-1}$ is positive definite, $y^T S_k^{-1} y \geq 0$, proving $M \succeq 0$. This guarantees $P_k \preceq P_{k-1}$, showing that the error covariance sequence decreases. If the system is stable, $P_k$ converges to a steady-state value.

Table 6: **Experiment hyperparameters.**

|  | Steps | Base model | CFG scale | Control strength |
|---|---|---|---|---|
| SDEdit | 50 | SD 1.4 | 4.0 | - |
| P2P | 50 | SD 1.4 | 7.5 | - |
| MasaCtrl | 50 | SD 1.4 | 7.5 | - |
| DDPM-Inv | 50 | SD 1.4 | 9 | - |
| RF-Edit | 28 | FLUX.1 dev | 2 | - |
| RF-Inversion | 28 | FLUX.1 dev | 3.5 | (0.7, 0.95) |
| FlowEdit | 28 | FLUX.1 dev | (1.5,5.5) | - |
| FlowChef | 28 | FLUX.1 dev | 2 | - |
| Ours | 28 | FLUX.1 dev | 3.5 | 0.95 |

# B  Additional details for experiment settings

First, we show the detailed hyperparameters setting for all baseline methods in Table 6. As demonstrated, for diffusion-based editing methods SDEdit [34] and P2P [16], we test both of them with 50 sampling steps and use StabeDiffusion 1.4 [41] as the base model. For all flow-based editing methods, we test all of them with 28 steps and use FLUX.1 dev [4] as base model. The timesteps are determined by Euler discrete scheduler. We also summarize the CFG scale and control strength in the table. These CFG values follow the default settings recommended in each baseline's implementation.

Next, we explain the hyperparameter settings for our proposed method. We set the steps to add the Kalman filter $l$ to 14, which is half of the total steps. And we set the steps $L$ to be the later half steps of the generation (i.e., steps 15 to 28). The hyperparameter $\delta$ determining the two types of measurement sequences is 6 by default. The coefficient hyperparameters $\mu$ and $\lambda$ are set to 0.7 and 0.1 by default.

We also set the matrices $A$, $B$, and $H$ in Eq. (9) to be identity matrices, which is computationally efficient and has proven to be effective in experiments. We also set the initial covariance matrix $P_0$ to be an identity matrix. The noise term is approximated by an identity matrix multiplied by a small coefficient 0.1 or 0.01. The complete procedure of our algorithm is given in Algorithm 2. All of our experiments are conducted on a single NVIDIA A40 GPU.

---

**Algorithm 2** Detailed procedure of Kalman-Edit

---

**Input:** original image $x_{\mathrm{s}}$, timesteps $\{t_i\}_{i=0}^{T}$, source prompt $c_s$, target prompt $c_t$, strength coefficients $\{\lambda\}_{i=0}^{T}$ and $\{\mu\}_{i=0}^{T}$, step sets $S_1$, $S_2$ for adding controller, and $L$ for adding Kalman filter (with $|L| = l$).
**Init:** $x_{t_N} \leftarrow x_{\mathrm{s}}, \quad M \leftarrow \emptyset$

/* Phase 1: Backward Denoising with Source Prompt */
**for** $i = N$ **to** $1$ **do**
$\quad v_{t_i} \leftarrow V_\theta(x_{t_i}, t_i, c_s)$
$\quad$ **if** $t_i \in S_1$ **then**
$\qquad x_{t_{i-1}} \leftarrow x_{t_i} + (t_{i-1} - t_i)\Big(\lambda\, v_{t_i} + (1 - \lambda)\, u_{t_i}\Big)$
$\quad$ **else**
$\qquad x_{t_{i-1}} \leftarrow x_{t_i} + (t_{i-1} - t_i)\, v_{t_i}$
$\quad$ **if** $t_i \in L$ **then**
$\qquad$ Add $t_i$ to $M$ $\hspace{3cm}$ $\triangleright$ Construct measurement sequence $\{y_i\}_{i=0}^{l}$
$x_{\mathrm{orig1}} \leftarrow x_{t_0}$

/* Phase 2: Forward Denoising with Target Prompt */
**for** $i = 0$ **to** $N - 1$ **do**
$\quad v_{t_i} \leftarrow -V_\theta(x_{t_i}, t_i, c_t)$
$\quad$ **if** $t_i \in S_1$ **then**
$\qquad x_{t_{i+1}} \leftarrow x_{t_i} + (t_i - t_{i+1})\Big(\lambda\, v_{t_i} + (1 - \lambda)\, u_{t_i}\Big)$
$\quad$ **else**
$\qquad x_{t_{i+1}} \leftarrow x_{t_i} + (t_i - t_{i+1})\, v_{t_i}$
$x_{\mathrm{mid}} \leftarrow x_{t_N}$ $\hspace{5cm}$ $\triangleright$ Middle latent

/* Phase 3: Backward Refinement with Controller $S_2$ */
**for** $i = N$ **to** $1$ **do**
$\quad v_{t_i} \leftarrow V_\theta(x_{t_i}, t_i, c_s)$
$\quad$ **if** $t_i \in S_2$ **then**
$\qquad x_{t_{i-1}} \leftarrow x_{t_i} + (t_{i-1} - t_i)\Big(\lambda\, v_{t_i} + (1 - \lambda)\, u_{t_i}\Big)$
$\quad$ **else**
$\qquad x_{t_{i-1}} \leftarrow x_{t_i} + (t_{i-1} - t_i)\, v_{t_i}$
$x_{\mathrm{orig2}} \leftarrow x_{t_0}$ $\hspace{4cm}$ $\triangleright$ Alternatively, one can use:
$\quad x_{\mathrm{orig2}} \leftarrow x_{\mathrm{orig1}} + (x_{\mathrm{mid}} - x_{\mathrm{s}})$

/* Phase 4: Forward Refinement with Target Prompt and Kalman Filter */
**for** $i = 0$ **to** $N - 1$ **do**
$\quad v_{t_i} \leftarrow -V_\theta(x_{t_i}, t_i, c_t)$
$\quad$ **if** $t_i \in S_2$ **then**
$\qquad x_{t_{i+1}} \leftarrow x_{t_i} + (t_i - t_{i+1})\Big(\lambda\, v_{t_i} + (1 - \lambda)\, u_{t_i}\Big)$
$\quad$ **else if** $t_i \in L$ **then**
$\qquad x_{t_{i+1}} \leftarrow x_{t_i} + (t_i - t_{i+1})\Big(\mu\, v_{t_i} + (1 - \mu)\, k_{t_i}\Big)$ $\triangleright$ $k_{t_i}$: Kalman filter terms (see Eq. (12))
$\quad$ **else**
$\qquad x_{t_{i+1}} \leftarrow x_{t_i} + (t_i - t_{i+1})\, v_{t_i}$
**Output:** edited image $x_{\mathrm{t}} \leftarrow x_{t_N}$

---

## C   Detailed Kalman-Edit algorithm

The detailed editing algorithm of our approach is presented in Algorithm 2. Phases 1 and 2 correspond to the first stage, where we construct the measurement sequence, while Phases 3 and 4 make up the second stage, where Kalman control is applied. Concretely, in Phase 1, we apply ODE inversion using rectified flow to construct the first part of the measurement sequence, $\{y_i\}_{i=0}^{\delta-1}$, from early inversion latents and obtain $x_{\text{orig1}}$. In Phase 2, we build the second part of the measurement sequence, $\{y_i\}_{i=\delta}^{l}$, using generation latents from later timesteps and acquire $x_{\text{mid}}$. We then compute $x_{\text{orig2}}$ either via shortcut estimation or a second inversion. In Phase 4, the Kalman filter is applied to produce more accurate results that align with both the target prompt and the structural integrity of the original image. For hyperparameters, we define filter strengths $\{\lambda\}_{i=0}^{T}$ and $\{\mu\}_{i=0}^{T}$ to control the guidance strength in the Kalman control process. We also specify step sets $S_1$ and $S_2$ to indicate where controllers should be applied, and step set $L$ to identify where the Kalman filter should be used. All hyperparameters are set to default values but may require tuning for different editing tasks.

## D   Illustration of more capabilities: Style transfer, Scene editing and large area modification

We showcase additional capabilities of Kalman-Edit, including style transfer, complex scene editing, and large-area modifications. Our method effectively adapts images to various styles, edits intricate backgrounds, and modifies extensive regions within an image. For instance, in Fig. 6, all four cases demonstrate our method's capability to edit complex background scenes while producing high-quality results. Figure 7 presents additional style transfer examples, such as converting a room into a medieval setting and performing large-area modifications, exemplified by the parrot case. In Fig. 8, we further highlight another large-area modification involving a giraffe. Moreover, Fig. 9, Fig. 10, and Fig. 11 illustrate various high-resolution style transfer cases, including transforming a picnic scene into a cartoon style and rendering a woman in the style of Van Gogh. Overall, these examples demonstrate the flexibility and effectiveness of our method in handling diverse and complex image editing tasks.

## E   Additional qualitative results

We present additional qualitative results, including real-world image edits and diverse editing tasks. We provide additional examples to further demonstrate the strong structural consistency and high editing quality achieved by our method. In Fig. 12, we present edited results on real-world images sampled from the DIV2K dataset. Our method effectively preserves the overall structure in non-target regions while accurately adhering to the editing prompts. For instance, in the boat insertion example, the foggy atmosphere is well preserved as a boat is naturally integrated into the scene. Other examples similarly exhibit high editing fidelity and strong structural consistency. Furthermore, Fig. 13 includes more diverse editing tasks, such as changing the breed of a dog. These results highlight the versatility and robustness of our approach in handling various types of image edits. In addition, Fig. 14 presents ablation studies on the effect of controller placement. The first row of images shows that applying controllers to early generation steps primarily influences the semantic structure, steering the output toward the original content (e.g., generating a cat) but missing background details. Conversely, the second row demonstrates that applying controllers to later steps refines visual details more effectively. These observations support our understanding of the controller's impact at different stages and help guide the selection of hyperparameters in Algorithm 2.

## F   Discussion of controller-based methods

In this section, we briefly discuss controller-based approaches, which are grounded in optimal control theory. As noted in [43], such methods can be applied to both diffusion-based and flow-based generative models (*i.e.*, they can operate on both stochastic differential equations (SDEs) and ordinary differential equations (ODEs)). With recent advances in flow-based models like Flux, there has been growing interest in applying control techniques specifically to ODE-based systems. From the perspective of optimal control theory, ODEs offer more favorable mathematical properties than SDEs, often enabling the derivation of more effective optimal controllers. For this reason, our proposed

| Original Image | FlowEdit | RF-Edit | Ours |
|:---:|:---:|:---:|:---:|

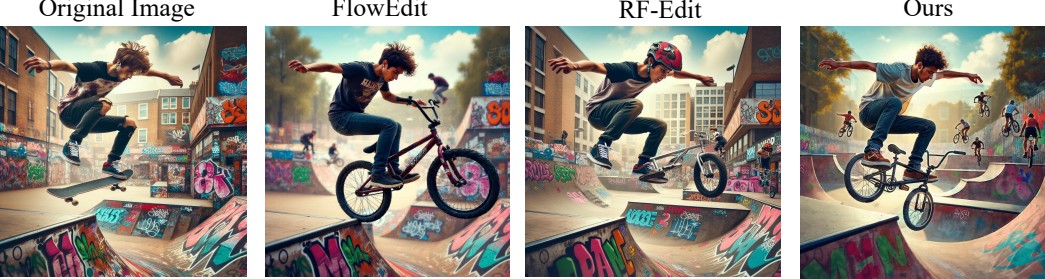

Change the skateboard to *a bike*, change the scene to *a skatepark*, and add *multiple riders performing tricks* in the background while maintaining the overall structure of the picture

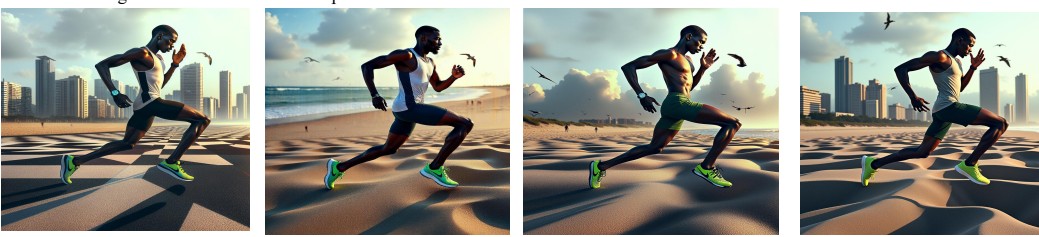

Change the *city track* to *a sandy beach environment with many skyscrapers and birds flying* in the background while maintaining the overall structure of the picture

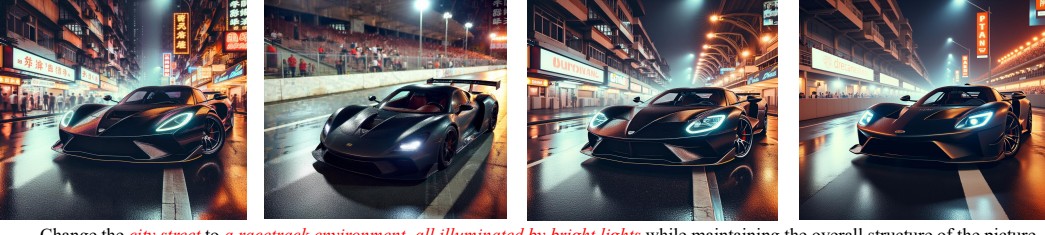

Change the *city street* to *a racetrack environment, all illuminated by bright lights* while maintaining the overall structure of the picture

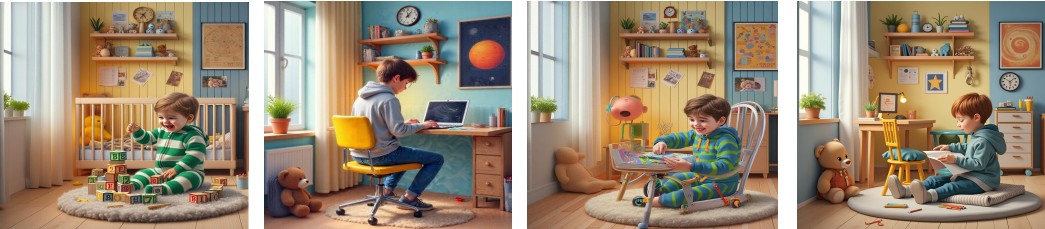

Change the child to *animate style wearing a grey hoodie and blue jeans, studying with books, and add a yellow chair and a teddy bear on the floor* while maintaining the overall structure of the picture

Figure 6: Illustration of intricate scene editing. We present complex scene editing cases, such as adding and modifying multiple room elements in the last-row example, and altering background scenes from a city track to a sandy beach and from a city street to a racetrack in the second-row example. Compared with baseline methods, our approach exhibits significantly stronger prompt adherence. These cases further demonstrate the robustness and versatility of our method in handling intricate editing tasks.

method adopts a flow-based framework. Nevertheless, similar to other controller-based approaches, our method can also be extended to diffusion-based models. Kalman-Edit presents a principled solution for achieving more accurate and consistent image editing through optimal control.

## G    Broader impact and limitations

**Broader impact.** While our Kalman-based method advances the quality and consistency of image editing methods, such techniques should be treated with caution due to their increasing potential for malicious use. Noteworthy, our method is training-free and does not rely on any private datasets for evaluation, thereby posing no data privacy concerns or associated negative impacts. To facilitate

| Original Image | FlowEdit | RF-Edit | Ours |
|:---:|:---:|:---:|:---:|
| 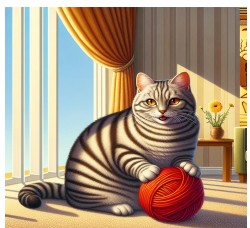 | 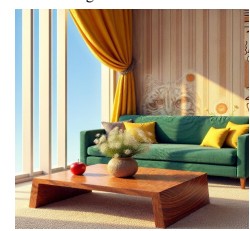 | 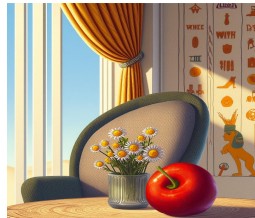 | 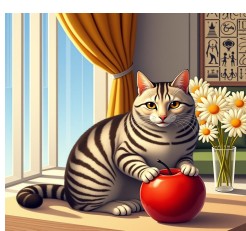 |

Change to *a rustic wooden table* in *a medieval-style kitchen with stone walls and antique kitchenware* while maintaining the overall structure of the picture

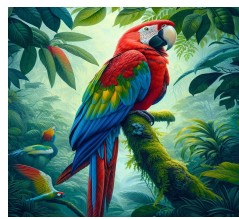 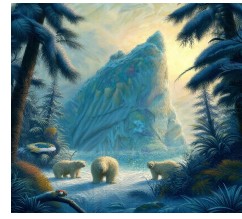 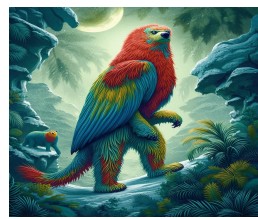 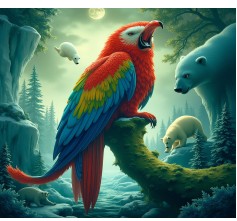

Add *an Egyptian hieroglyphic mural with symbols on the wall, a green couch, a wooden coffee table with a vase of daisies and a red apple on it* while maintaining the overall structure of the picture

Change to *a snowy arctic landscape* with *icy cliffs, snow-covered trees, and polar bears* while maintaining the overall structure of the picture

Figure 7: Illustration of large-area modification. We present examples including room style transformation (first-row case) and the addition of multiple new elements (second-row case). Furthermore, we demonstrate a case involving substantial background modification, where a forest scene is replaced with icy cliffs and polar bears (last-row case).

further open research into its practical uses and any potential societal impacts, our code would be open sourced at https://github.com/anonymous-138384/Kalman-Edit-Pytorch/.

**Limitation.** Due to our design focus on leveraging historical inversion latents as the measurement sequence to rectify the final generation process, the method requires additional hyperparameter tuning for tasks such as object removal. We illustrate these failure cases in Fig. 15. In addition, the noise and artifacts from the original image tend to appear in the edited images at our default control strength.

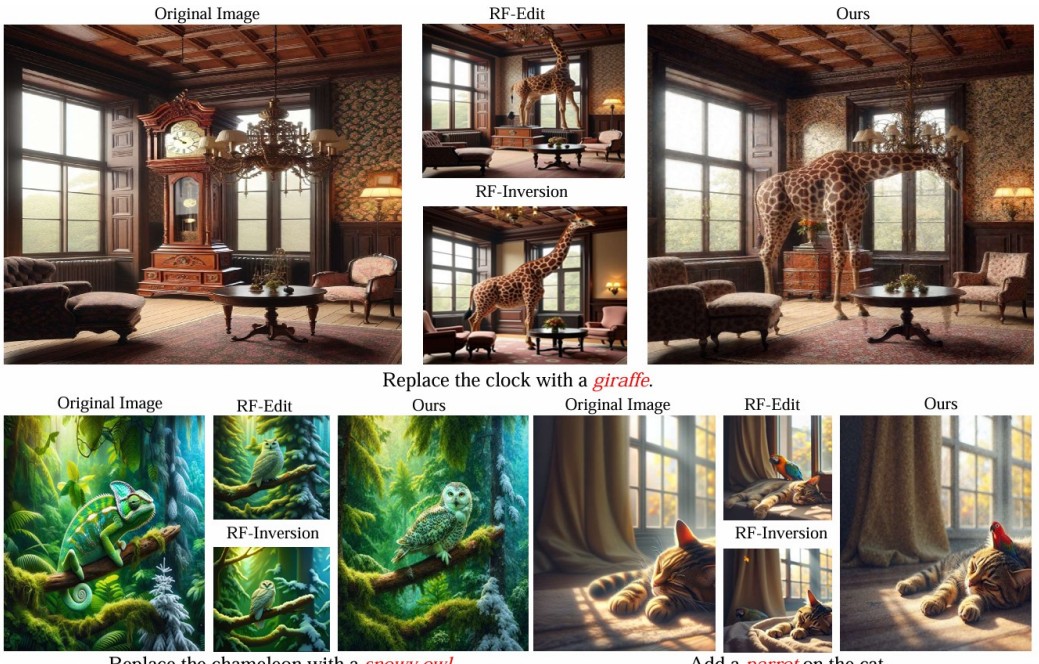

Figure 8: Illustration of large-area modification and complex scene editing. The giraffe example demonstrates our method's ability to preserve image structure, as evidenced by the chandelier remaining correctly positioned. The owl example features a complex background, and our result retains most structural details in the non-target regions. In the cat example, our method maintains strong structural consistency while accurately adhering to the target prompt.

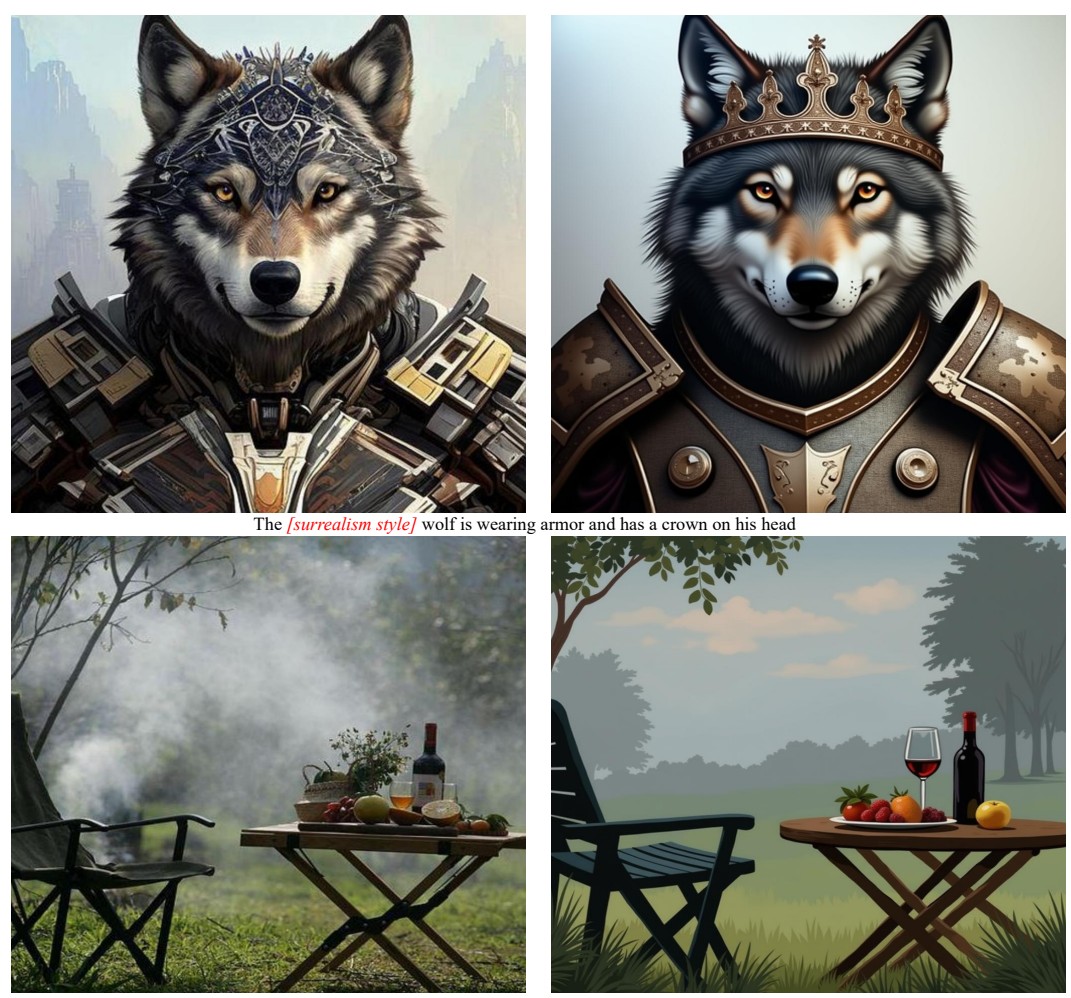

The *[surrealism style]* wolf is wearing armor and has a crown on his head

*[Cartoon style]* a picnic table with a bottle of wine and fruit on it

Figure 9: Additional qualitative results on the style transfer task. We present examples of style transfer in surrealism and cartoon styles.

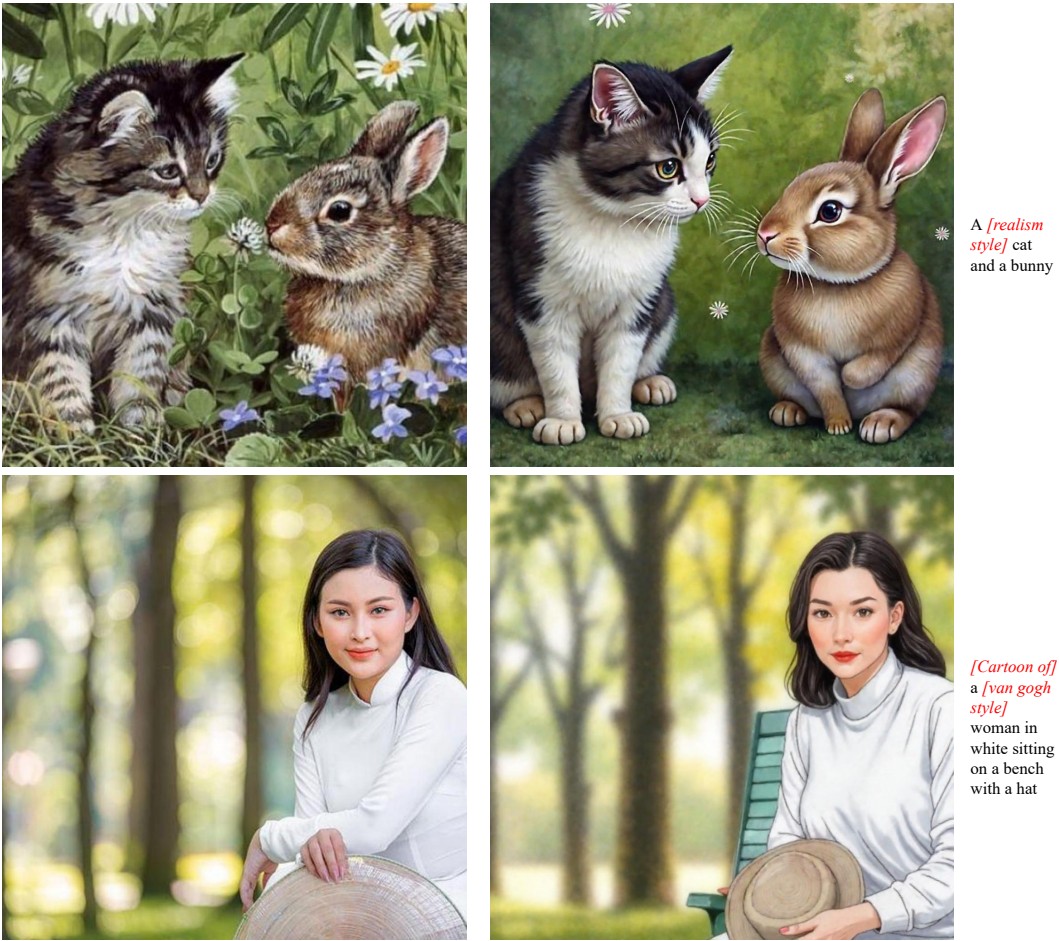

A *[realism style]* cat and a bunny

*[Cartoon of]* a *[van gogh style]* woman in white sitting on a bench with a hat

Figure 10: Additional qualitative results on the style transfer task. We present examples of style transfer in realism and van gogh styles.

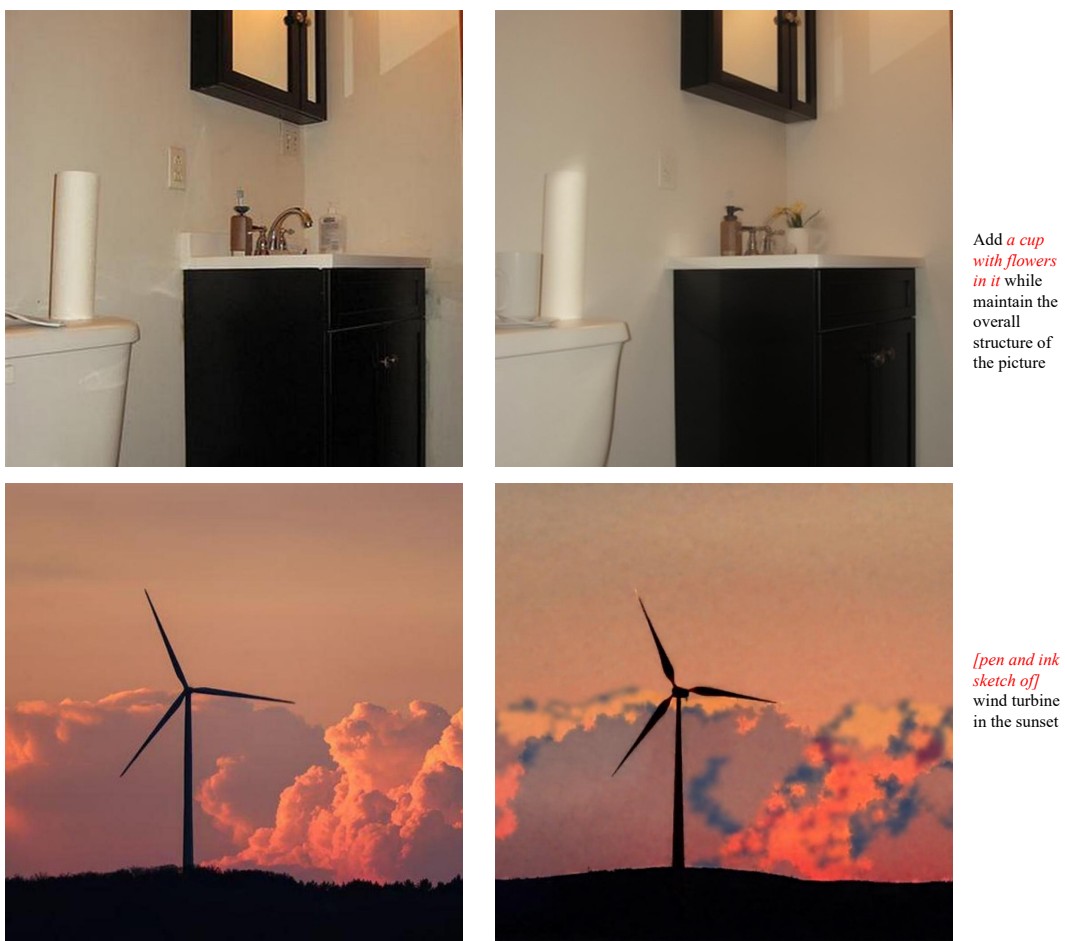

Add *a cup with flowers in it* while maintain the overall structure of the picture

*[pen and ink sketch of]* wind turbine in the sunset

Figure 11: Additional qualitative results on the style transfer task. We present an example of style transfer to a pen-and-ink style, and additionally include a bathroom editing case.

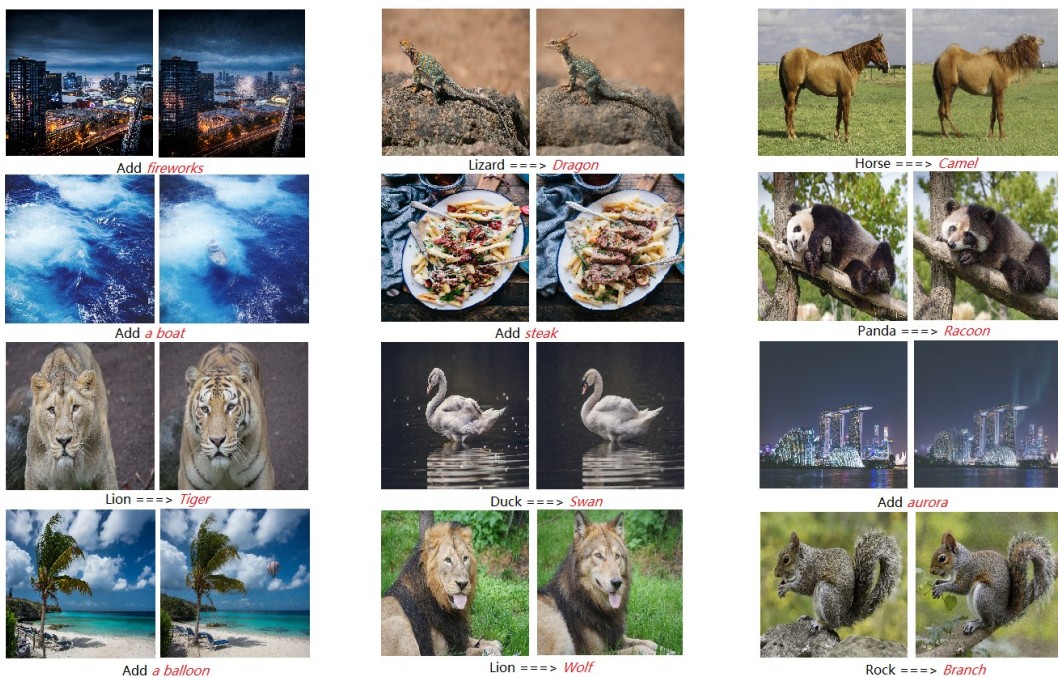

Figure 12: Additional results on real-world image editing. We present examples involving complex scenes such as urban environments and beaches, as well as images with intricate structures like the boat and the panda. These results highlight the flexibility and effectiveness of our method in handling diverse and challenging editing tasks.

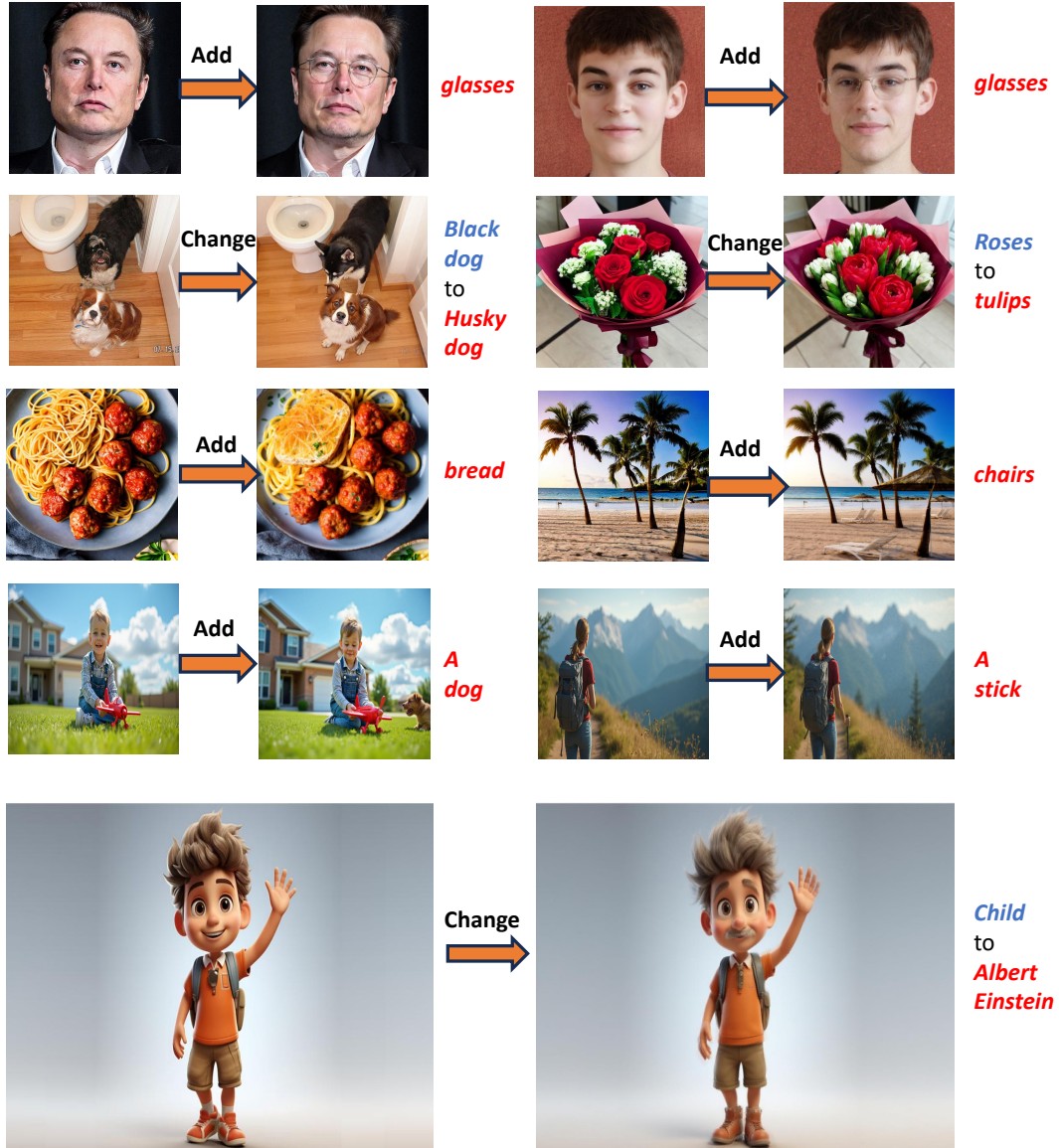

Figure 13: Additional results on diverse image editing tasks. We present more examples with a wide range of target prompts, such as changing a dog's breed and transforming roses into tulips. These results further demonstrate the strong performance and versatility of our approach.

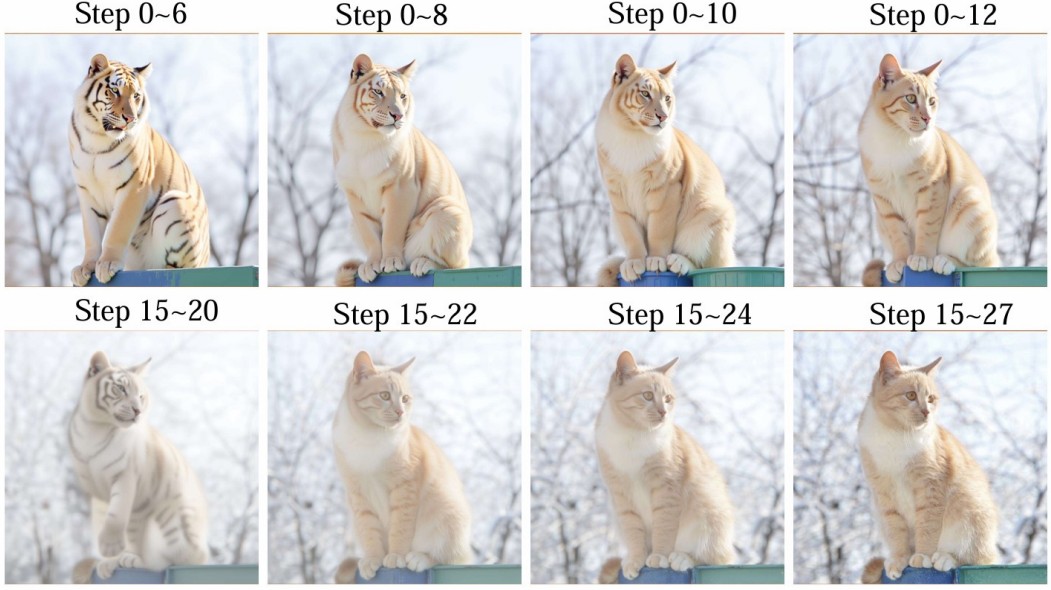

Figure 14: Additional ablation results on controller placement. We evaluate the effects of adding controllers at different stages of the generation process. These insights inform the hyperparameter choices in our proposed algorithm. For a detailed analysis, please refer to Appendix E.

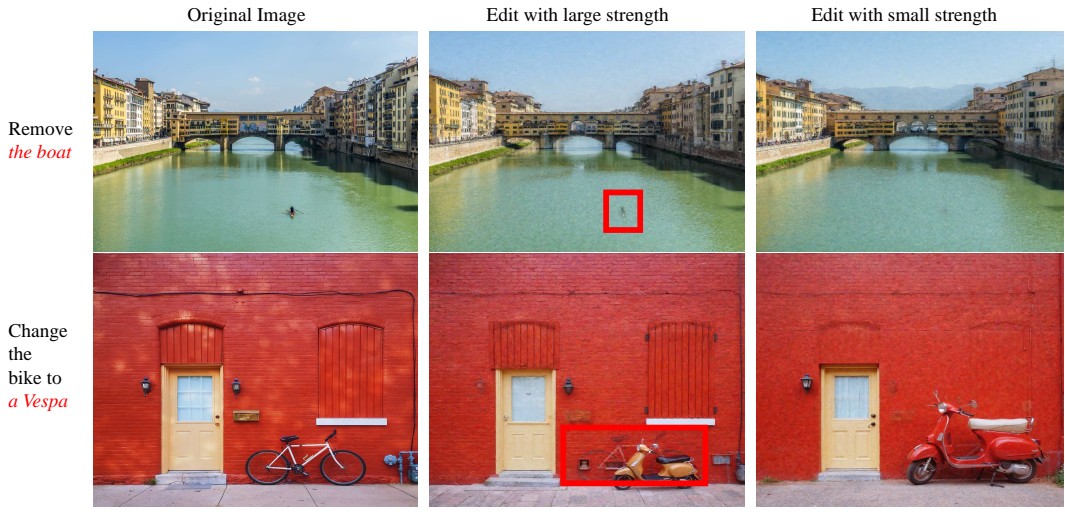

Figure 15: Limitation of our method. Improper hyperparameter settings may cause our method to fail in some editing cases, such as certain object removal tasks. See Appendix G for detailed explanation.

