# OpenReview forum: "Enhancing Consistency of Flow-Based Image Editing through Kalman Control"
_NeurIPS.cc/2025/Conference — NeurIPS 2025 poster_

### Official Review · Reviewer_ndBN · 2025-07-02

**Clarity:** 2
**Significance:** 2
**Originality:** 3
**Rating:** 4
**Confidence:** 4

**Summary:**

Flow-based image editing suffers from structural inconsistencies in non-target areas, due to the accumulation of small errors. To address this, the editing process is formulated as an LQG control problem, and Kalman-Edit is introduced, which integrates a Kalman filter to exploit historical inversion trajectories for preserving fine details. Additionally, it proposes a faster variant, Kalman-Edit*, that approximates the second inversion step to reduce computation while incurring only minimal quality loss.

**Questions:**

- Why does the proposed method require a two-stage approach? Section 3.3 (lines 146–153) offers some explanation, but it remains unclear why a single-stage Kalman control fails while a two-stage approach succeeds. First, selecting appropriate timesteps for Kalman filtering is inherently challenging, and the proposed method still relies on selected ranges of timesteps. Second, the authors claim that applying Kalman control in one inversion and forward pass causes blurring and artifacts, yet they do not explain why the filter in stage 2 avoids these issues. An ablation study directly comparing a single-stage Kalman control setup with the two-stage pipeline would help clarify this choice.

- Unlike other methods, the proposed approach requires two generation passes, even with the approximate second inversion in Kalman-Edit*. Therefore, comparison of inference times against competing techniques is necessary.

- Tables 1 and 2 only include flow-based baselines, and the qualitative results omit any diffusion-based methods. Since diffusion models are inherently slower, demonstrating comparable performance between the proposed method and diffusion-based approaches would underscore the efficiency of the proposed framework.

- What inversion method do P2P and MasaCtrl use? DDIM inversion yields poor reconstruction quality when editing real images. Both could potentially benefit from more advanced inversion schemes, such as Direct Inversion [1], NMG [2], or SPDInv [3]. It would strengthen the evaluation to compare against these advanced methods.

*Minor Suggestions*
- Provide additional implementation details: how are editing prompts formulated for the DIV2K dataset, and which timesteps are used for S1 and S2 in Algorithm 2?


[1] Ju, X., Zeng, A., Bian, Y., Liu, S., & Xu, Q. Direct inversion: Boosting diffusion-based editing with 3 lines of code. ICLR 2024.
[2] Cho, H., Lee, J., Kim, S. B., Oh, T. H., & Jeong, Y. (2024). Noise map guidance: Inversion with spatial context for real image editing. ICLR 2024.
[3] Li, R., Li, R., Guo, S., & Zhang, L. (2024, September). Source prompt disentangled inversion for boosting image editability with diffusion models. ECCV 2024.

**Ethical Concerns:**

["NO or VERY MINOR ethics concerns only"]

**Final Justification:**

The author's response resolve my concern, so I raise my score accordingly

**Limitations:**

yes

**Paper Formatting Concerns:**

No concern

**Quality:**

3

**Strengths And Weaknesses:**

**[Strengths]**
- Reformulates the editing process as an LQG control problem and integrates a Kalman filter to improve structural consistency.
- Proposes an accelerated variant, Kalman-Edit*, which omits a second inversion to cut computation time while maintaining high-quality results.

**[Weaknesses]**
- The motivation for the two-stage design is not thoroughly explained.
- Evaluation is limited, relying mainly on comparisons with flow-based editing baselines.

---

> ### Author Rebuttal · Authors · 2025-07-27
>
> We greatly appreciate your constructive comments. Below, we provide our responses to the concerns raised.
>
> [W1] Motivation for the two-stage design
> - This design is motivated by the observation that previous controller-based methods have difficulty preserving editing consistency. And we find that harnessing history trajectory could mitigate this issue, which naturally aligns with Kalman control. Therefore, we propose the two-stage method: stage 1 corresponds to the controller-based method that collects the history trajectory, while stage 2 corresponds to Kalman control utilizing that history trajectory. The two-stage approach better utilizes the structural information in the trajectory and enhances the editing consistency.
>
> [Q1] Necessity of the two-stage design
> - The necessity of the two-stage design arises from the editing dilemma as presented in Figure 1. Figure 1.a shows that previous controller-based methods tend to reach under-editing or over-editing area and lead to editing failure. With the two-stage approach, we can better utilize the structural information in the history trajectory with the help of the Kalman filter and succeed in editing, as shown in Figure 1.b. Moreover, we have conducted the ablation study comparing vanilla controller-based methods and our method with Kalman filter added.  Kalman filter significantly improves the performance of the controller-based method, demonstrating the effectiveness of our two-stage design. Please refer to Table 5 in the Experiment section for more details.
>
> [Q2] Time cost comparison
> - We have presented the running time comparison of different methods. Our proposed Kalman-Edit* is slower than RF-Inversion and Flowchef, but faster than all other baselines. This shows that our approach strikes a competitive balance between editing performance and efficiency. Please refer to Figure 4 in the paper for more details.
>
> [W2/Q3/Q4] Quantitative comparison with more recent diffusion-based methods
> - We conduct additional experiments on the three listed diffusion-based methods. The results are shown in the two tables below. Our method still shows better performance across most metrics on SFHQ and HQ datasets compared to more advanced inversion schemes Direct Inversion [1], NMG [2], and SPDInv [3]. This shows the effectiveness of our approach.
>
> **Quantitative comparison on the SFHQ dataset**
> | Method            | Face Rec. ↓ | CLIP-I ↑   | LPIPS ↓    | CLIP-T ↑   | DreamSim ↓ |
> | ----------------- | ----------- | ---------- | ---------- | ---------- | ---------- |
> | **NMG**           | 0.4437      | 0.8264     | 0.1906     | 0.2820     | 0.1825     |
> | **Direct-Inv**    | 0.4276      | 0.8350     | 0.1703     | 0.2686     | 0.1535     |
> | **SPD-Inv**       | 0.4625      | 0.8561     | 0.1618     | 0.2646     | 0.1480     |
> | **RF-Edit**       | 0.4051      | 0.8984     | 0.1562     | 0.2910     | 0.1591     |
> | **RF-Inversion**  | 0.4325      | 0.8927     | 0.1720     | **0.3012** | 0.1889     |
> | **FlowEdit**      | 0.4856      | 0.8579     | 0.1687     | 0.2905     | 0.2375     |
> | **FlowChef**      | 0.4013      | 0.8769     | 0.1401     | 0.2832     | 0.1487     |
> | **Kalman-Edit**   | **0.3958**  | **0.9167** | **0.1332** | 0.2921     | **0.1408** |
> | **Kalman-Edit**\* | 0.4696      | 0.8871     | 0.1892     | 0.2936     | 0.2227     |
>
> **Quantitative comparison on the HQ dataset**
> | Method            | CLIP-T ↑   | CLIP-I ↑   | LPIPS ↓    | DINO ↑     | DreamSim ↓ |
> | ----------------- | ---------- | ---------- | ---------- | ---------- | ---------- |
> | **NMG**           | 0.1879     | 0.8916     | 0.2826     | 0.8310     | 0.1672     |
> | **Direct-Inv**    | 0.1676     | 0.9153     | 0.2866     | 0.8188     | 0.1834     |
> | **SPD-Inv**       | 0.1864     | **0.9221** | 0.2756     | **0.8237** | 0.1548     |
> | **RF-Edit**       | 0.1842     | 0.9141     | 0.2383     | 0.8197     | 0.1492     |
> | **RF-Inversion**  | 0.1825     | 0.9033     | 0.3074     | 0.7963     | 0.1662     |
> | **FlowEdit**      | 0.1877     | 0.8813     | 0.2846     | 0.7467     | 0.2238     |
> | **FlowChef**      | 0.1928     | 0.9023     | 0.2925     | 0.8053     | 0.1537     |
> | **Kalman-Edit**   | **0.1943** | 0.9062     | **0.2345** | 0.7929     | **0.1353** |
> | **Kalman-Edit**\* | 0.1870     | 0.8696     | 0.3615     | 0.7123     | 0.2276     |
>
> ---
> [1] Ju, X., Zeng, A., Bian, Y., Liu, S., & Xu, Q. Direct inversion: Boosting diffusion-based editing with 3 lines of code. ICLR 2024.
>
> [2] Cho, H., Lee, J., Kim, S. B., Oh, T. H., & Jeong, Y. (2024). Noise map guidance: Inversion with spatial context for real image editing. ICLR 2024.
>
> [3] Li, R., Li, R., Guo, S., & Zhang, L. (2024, September). Source prompt disentangled inversion for boosting image editability with diffusion models. ECCV 2024.

---

> > ### Comment · Reviewer_ndBN · 2025-08-05
> >
> > Thank you for your thorough rebuttal. It has addressed my concerns, and I will raise my score accordingly.

---

### Official Review · Reviewer_bsYB · 2025-07-02

**Clarity:** 3
**Significance:** 3
**Originality:** 3
**Rating:** 3
**Confidence:** 3

**Summary:**

This paper formulates the image editing task with *Rectified flow* model as a control problem and proposes *Kalman-Edit*, which uses *Kalman filter* to leverage the historical editing trajectory. For enhancing image consistency during the editing process, *Kalman-Edit* reuses early-stage representation from the overall sampling trajectory. Through this work, the authors aim to address the issues of structural consistency and editing quality inherent in previous approaches. The authors conducted extensive experiments on four datasets—SFHQ, HQ, ZONE, and DIV2K—to evaluate the performance of the proposed method.

**Questions:**

Q1: In Section 3.3, the “two-stage image editing” approach is introduced, but no ablation study is provided for it. Why was an ablation analysis for this component omitted, and why is there no ablation on the number of stages?

Q2: In the RF-Edit, video editing is demonstrated; could the same method be applied to video editing tasks as well?

**Ethical Concerns:**

["NO or VERY MINOR ethics concerns only"]

**Limitations:**

yes

**Paper Formatting Concerns:**

No major paper formatting concerns

**Quality:**

2

**Strengths And Weaknesses:**

**Strengths**
1. The authors clearly identified the problems using rectified flow models for editing and articulated the motivation with clarity.
2. This paper establishes a link between image editing and control theory, introducing a novel approach that leverages the Kalman filter.
3. Extensive experiments were conducted across multiple datasets, providing the evaluation of the proposed approach’s performance.

**Weaknesses**
1. The qualitative results lack a sufficient set of baseline comparison models. In addition to *RF-Edit* and *RF-Inversion*, visualizations comparing against *SDEdit*, *P2P*, *MasaCtrl*, and *DDPM-Inv*—which the authors referenced in their quantitative results—are also necessary. Additionally, visual results for *Kalman-Edit*, one of the paper’s contributions, are needed.
2. The baseline comparison experiments focus solely on changing or adding specific objects; additional comparisons on image stylization are necessary.
3. Overall, the qualitative results are not good. In *Figure 2*, the reported edits involve tiny objects or mere texture changes, and the output quality appears poor (e.g., plant → flower, wall → brick walls). In terms of consistency, RF-Inversion preserves the original image far better in *Figure 3*. Moreover, in *Figure 5* cartoon-style transformation, a teddy bear is added unexpectedly; in *Figure 6* snow-covered tree addition, the image fails to reflect the change properly; in the two-dimensional Egyptian style transformation, there is no proper alteration; and cup addition in *Figure 9*, the image does not show the cup effectively—suggesting that the proposed method’s performance is not good and robust.
4. The images in the qualitative results are presented at low resolution, making it difficult to tell which model performs better, even when zoomed in. Uploading higher-resolution images would help readers understand the performance of the proposed method more effectively.

---

> ### Author Rebuttal · Authors · 2025-07-27
>
> We are grateful for your thoughtful suggestions and respectfully address your primary concerns below:
>
> [W1/W2] Lacking diffusion-based qualitative comparison and image stylization results
> - We will add an additional visual comparison between diffusion-based methods and ours. Since we can not upload the visualization result here, we conduct an additional quantitative experiment on the PIE benchmark [1] to show the effectiveness of our method in the image stylization task. Our method owns superior performance on the image stylization task across different metrics, compared to both diffusion-based and flow-based methods.
>
> [W3] Concern about editing quality
> - For Figure 2, our approach achieves good editing quality and maintains strong structural consistency. To verify this, we conduct an additional experiment to evaluate the local area similarity. As shown in the table below, our approach achieves the best similarity results across all evaluation metrics. This demonstrates the strong structural consistency of our method. Some of the changes in Figure 2 may appear minor because the prompt specifically targets modifications in a small local area. To better illustrate the results and demonstrate the effectiveness of our method, we will add more high-resolution image results to the Appendix.
> - For Figure 3, we would like to highlight that our approach preserves more face structural consistency and has higher face similarity. We qualitatively evaluate the average Face Rec metric of the images in Figure 3. Our results have 0.4021 average Face Rec, which is lower than RF-Inversion with 0.4213 average Face Rec. This shows that our method preserves better facial consistency.
> - For Figures 5 and 6, all the images are from the HQ dataset and edited with long and complex prompts. In both Figures, we only sketch the main editing area, which does not reflect the whole target prompt. We are sorry about the confusion. For example, the teddy bear case is edited with the target prompt “A teenager in a grey hoodie and blue jeans is sitting on a yellow chair at a wooden desk studying from a book and using a laptop. The room has blue walls, a shelf with books and a clock, a large framed space poster on the wall, and a teddy bear on the floor next to a backpack and scattered school supplies.” Therefore, the teddy bear appears in the target prompt and should be included in the edited image as expected. We will add the whole target prompts of each editing case to the Appendix.
> - For Figure 9, most cases can reflect the effectiveness of our approach. Regarding the case of the mentioned cup, since the editing area is very small and the image is in low resolution, it does not show up effectively. We sincerely thank you for the reminder and would add high-resolution cases here.
>
> [W4] Lacking high-resolution images
> - We will add higher-resolution images to our paper. Thank you for your valuable advice.
>
> [Q1] Lacking ablation study results
> - We have conducted the ablation study between the previous one-stage controller-based method (i.e. w./o. Kalman filter) and our two-stage approach (i.e. w./ Kalman filter) in our paper. The ablation results demonstrate the effectiveness of our approach. Please refer to Table 5 for more details and Section 4.2 for more analysis.
>
> [Q2] Concern about extension to video editing
> - Our work focuses on the image editing task. The effectiveness of our approach for video editing remains to be explored. Compared to RF-Edit, our method does not require the modification of attention values and is expected to have better video editing flexibility and quality. We would try to extend our method to video editing in future work.
>
> **Quantitative comparison on the PIE benchmark image stylization task**
> | Method            | CLIP-T ↑   | DreamSim ↓ | CLIP-I ↑   | LPIPS ↓    |
> | ----------------- | ---------- | ---------- | ---------- | ---------- |
> | **SDEdit**           | 0.2842     | 0.1124     | 0.9230     | 0.2575     |
> | **P2P**    | 0.2988     | 0.1010     | 0.9375     | 0.1691     |
> | **MasaCtrl**       | 0.3041     | 0.1504     | 0.9150     | 0.1607     |
> | **DDPM-Inv**       | 0.2913     | 0.1103     | 0.9204     | 0.1461     |
> | **RF-Edit**       | 0.2845     | 0.1212     | 0.9271     | 0.1592     |
> | **FlowEdit**      | 0.2876     | 0.1686     | 0.8802     | 0.1623     |
> | **Kalman-Edit**   | 0.2903     | **0.0912** | **0.9451** | **0.1402** |
> | **Kalman-Edit**\* | **0.3135** | 0.1347     | 0.9381     | 0.1652     |
>
> **Quantitative evaluation of local area similarity in Figure 2**
> | Method            | DreamSim ↓ | CLIP-I ↑   | LPIPS ↓    |
> | ----------------- | ---------- | ---------- | ---------- |
> | **RF-Inversion**   | 0.3322     | 0.8179     | 0.2128     |
> | **RF-Edit**       | 0.2646     | 0.8467     | 0.1546     |
> | **Kalman-Edit** | **0.2318** | **0.8640** | **0.1127** |
>
>
> ---
> [1] Ju, X., Zeng, A., Bian, Y., Liu, S., & Xu, Q. (2024). PnP Inversion: Boosting Diffusion-based Editing with 3 Lines of Code. ICLR 2024.

---

> ### Comment · Reviewer_bsYB · 2025-08-05
>
> Thanks to the authors for the additional evaluations and thoughtful responses. However, concerns regarding **W3** and **qualitative results** remain unresolved.
>
> The claim that the model shows strong structural consistency may not be convincing. In many examples, only small parts of the image are edited, or only textures are changed. These cases naturally lead to high consistency, without proving the model’s robustness. In Figure 2, most edits are minor or limited to small regions, which supports this concern.
>
> In Figure 5, the issue raised about the top-left image (changing child to boy) was said to be a misunderstanding due to the missing full prompt. However, even with the complete prompt—“A teenager in a grey hoodie and blue jeans is sitting on a yellow chair”—the output does not match the description. The boy is not sitting on a chair and is not wearing a grey hoodie. The claimed cartoon-style transformation is also unclear. The before-and-after images look almost the same in terms of style.
>
> The concern regarding Figure 6 was previously mentioned in the earlier review and remains unresolved. In particular, the cat and parrot images are still not properly reflected target prompt (Egyptian style, snowy landscape, ...) in the edited results, which further reinforces the concern.
>
> In image editing, qualitative results are just as important as quantitative scores. Although the paper reports good quantitative performance, the qualitative results are not convincing. Most edits only add or change small objects or textures, and the images are shown at low resolution. As a result, it is difficult to clearly evaluate the method’s robustness from the figures provided.

---

> > ### Author Response · Authors · 2025-08-05
> > **Reply to Reviewer bsYB**
> >
> > Thank you for your constructive feedback. We address your concerns as follows:
> >
> > Q1: Most edits are minor or limited to small regions
> > * Please see Figure 5 with several scene editing cases that modifies a large region (>40%). In the top-right example, our method modifies the whole island while adding an additional bridge. This demonstrates the capability of our method in major edits.
> >
> > Q2: Prompt adherence
> > * Our method already exhibits better prompt alignment than baselines. We test all baselines on Figure 5 and 6, and find that our method prevails in successfully modifying elements like yellow chair, wooden desk and teddy bear in the top-left example of Figure 5, as well as the Egyptian mural and polar bears in the cat and parrot cases of Figure 6, while all other baselines fail. Moreover, our approach demonstrates superior structural consistency, as seen in the preservation of the athlete’s pose in the bottom-right image of Figure 5, which is not well maintained by other methods. We will include these comparisons in the revision, and please refer to Figures 2-3 and Tables 1-3 in the main paper for further quantitative and qualitative evidence of our method outperforming baselines .
> > * The editing cases in Figure 5 and 6 have long and complicated prompts, making all methods hard to accomplish well. This can be reflected by the low CLIP-T score in the HQ dataset shown in Table 2. Compared to other baselines, our method shows superior prompt adherence, which can be reflected in the CLIP-T score. We will include the editing results of other baselines for better visualization comparison.
> > * For the editing failures, we attribute them to the prompting following capability of base diffusion model itself (FLUX.1 dev). We test similar prompts in text-to-image generation with current base model and found it fails to generate fine-grained structral details according to the prompts. This will be resolved as the base model continues to improve.

---

> ### Comment · Reviewer_bsYB · 2025-08-09
>
> Thanks to the authors for the detailed responses.
>
> However, my concern that most edits are minor or confined to small regions remains unresolved. While the top-right example in Figure 5 indeed shows a large-region edit as claimed, the majority of examples in the main paper and the supplementary material continue to illustrate small, local edits. Regarding the qualitative comparisons on the ZONE and DIV2K datasets, the evaluation covers only two baselines and four scenes (Figure 2), which is too limited to support broad claims. For Figures 5 and 6, the authors state that the baseline models failed; however, no side-by-side qualitative comparisons are provided. Consequently, it is not possible for a reviewer to verify these claims. In Figure 5, the title of the figure describes a style-transfer example, but the bottom-left image appears to add an Egyptian mural to the background rather than transforming the entire image into an Egyptian style, which creates confusion between the description and the actual result.
>
> Even if space constraints prevented extensive comparisons in the main paper, the additional results section—where space is typically less constrained—should have included more comprehensive qualitative comparisons. Moreover, many visual results are uploaded at low resolution; in some cases (e.g., Figure 9, “adding cup”), the images are degraded to the point that it is difficult to judge the edits. Since the central task is image editing, low-resolution visuals hinder a fair assessment and are not recommended.
>
> Taking these points together, my concerns about the qualitative results remain unresolved. Substantial revisions would be necessary to address them, and in the paper’s current state, I cannot recommend acceptance. I regrettably recommend rejection, as I believe this needs another round of reviewing, where reviewers can inspect the visual comparison with clear and more qualitative results.

---

> ### Author Response · Authors · 2025-08-09
> **Reply to Reviewer bsYB**
>
> Thank you for your constructive feedback.
>
> Due to this year’s review policy, we are not permitted to upload additional qualitative results at this stage.
> We have already provided extensive quantitative comparisons demonstrating our model’s superior performance. We have also included many qualitative results, including the comparisons in Figures 2 and 3 in the main paper,  large area modification cases in Figures 5 and 6, additional comparison cases in Figure 7, as well as real-world editing cases in Figures 8 and 9. In addition, we would love to include more comprehensive qualitative comparisons and higher-resolution images in the revision. Specifically, we will:
> * Provide visualization comparisons for each case in Figures 5 and 6 in the Appendix, highlighting our model’s superior editing capabilities.
> * Add more editing cases involving large-area modifications, similar to those in Figures 5 and 6, and compare them against all other baselines.
> * Upload higher-resolution results for each case in Figure 9, along with additional editing cases using a broader variety of prompt types.
>
> We appreciate your valuable feedback again and hope this helps address your concerns.

---

### Official Review · Reviewer_YwVg · 2025-07-03

**Clarity:** 3
**Significance:** 3
**Originality:** 3
**Rating:** 4
**Confidence:** 2

**Summary:**

This paper introduces a new method to enhance the consistency and relevance of image editing problem. The proposed method exploits Kalman control concept in the flow-matching equation and optimize the image editing into 2 stages. It starts with a inversion process to get the noise from the input image and then followed by a Generation process to produce the target image. The proposed method also provide a faster version to skip the second inversion process with approximation to accelerate the image generation throughput. Extensive experiments on qualitative and quantitative evaluation demonstrate the effectiveness of the proposed method.

**Questions:**

This paper proposes an efficient and performing method to control the feature editing using Kalman-control with strong solid formulation and reasonable assumptions. The authors are suggested to address minor issues raised in the weakness during rebuttal period.

**Ethical Concerns:**

["NO or VERY MINOR ethics concerns only"]

**Final Justification:**

I would like to keep my initial ratings as accept because this paper proposes a novel theory to do flow matching using kalman filtering and comes up with good experiment results to demonstrate the effectiveness of the theory. The rebuttal messages and additional experiment results have addressed the previous concerns.

**Limitations:**

yes, with failure cases and limitation in supplementary material.

**Paper Formatting Concerns:**

No concerns for formmating

**Quality:**

3

**Strengths And Weaknesses:**

### Strength
1. The qualitative experiments show that the proposed method is really efficient and performing. It keeps the identity of the original person very well.
2. The proposed method comes with a verifiable code implementation for results reproducing.
3. The proposed method is technically sound with equation derivation and proofs.
4. Extensive experiments show that the proposed method outperforms the baseline state of the art methods significantly and consistently on the benchmarking datasets.

### Weakness
1. What is the reason that Kalman-Edit* shortcut has significant higher performance on CLIP-T compared to Kalman-Edit in Table 3 ?

---

> ### Author Rebuttal · Authors · 2025-07-27
>
> We sincerely appreciate your valuable feedback. Our response is as follows.
>
> [W/Q] Reason of Kalman-Edit* owning higher CLIP-T performance
>
> - Kalman-Edit* achieves significantly higher performance on CLIP-T in Table 3, primarily because the evaluation datasets, Zone and DIV2K, are dominated by editing tasks involving local modifications like object addition or replacement. Though the second-pass inversion utilizes more structural information, it is more constrained and lacks flexibility when tackling small area modifications.
> - By contrast, the shortcut estimation in Kalman-Edit* introduces greater flexibility in adhering to the target prompt as it provides a looser estimation method that utilizes a first-order approximation in the vector field. When handling the local modifications task, this method would yield a relatively small estimation error. To verify this point, we conduct an additional quantitative experiment showing the local area similarity. As shown in the table below, our approach achieves superior performance across all evaluation metrics, showing that our approach has relatively minor errors in estimating the local area in different images. This leads to better target prompt adherence, as many prompts in the evaluation datasets contain local area descriptions. As a result, Kalman-Edit* demonstrates superior performance on the CLIP-T metric.
>
> **Quantitative evaluation of local area similarity on Zone dataset**
> | Method            | DreamSim ↓ | CLIP-I ↑   | LPIPS ↓    |
> | ----------------- | ---------- | ---------- | ---------- |
> | **RF-Inversion**   | 0.2172     | 0.8558     | 0.1618     |
> | **RF-Edit**       | 0.1743     | 0.8831     | 0.1473     |
> | **Kalman-Edit**\* | **0.1508** | **0.9012** | **0.1322** |

---

### Official Review · Reviewer_7uLu · 2025-07-15

**Clarity:** 3
**Significance:** 3
**Originality:** 3
**Rating:** 5
**Confidence:** 3

**Summary:**

This paper formulates image editing as an optimal control problem, aiming to satisfy the target condition while minimizing editing cost. It introduces the classical Kalman filter into the image editing pipeline to better preserve details from the original image. The method is seamlessly integrated into the standard forward and inverse processes of diffusion models. Additionally, the authors propose a shortcut technique based on approximate vector field velocity estimation, which enhances editing quality without incurring significant computational overhead.

**Questions:**

How sensitive are the final results to different initializations or hyperparameter settings of the Kalman filter?

**Ethical Concerns:**

["NO or VERY MINOR ethics concerns only"]

**Final Justification:**

After reading comments from authors and other reviewers, I maintain my rating for acceptance. I believe the value of this paper lies in connecting a theory-grounded control literature to the task of image editing. I agree with bsYB that this paper should be improved with better visual results and more quantitative evaluation. Hoping the authors could improve on those in the final version.

**Limitations:**

Authors discuss the limitations in the final chapters and this paper has no potential negative societal impact.

**Paper Formatting Concerns:**

No formatting concerns.

**Quality:**

3

**Strengths And Weaknesses:**

Strengths:

(1) The paper presents a compelling perspective by framing image editing as a control problem and effectively incorporates the classical Kalman filter to demonstrate tangible improvements.

(2) The motivation is well-articulated, and the proposed method achieves significant performance gains.

(3) The approach is firmly grounded in optimal control theory and is supported by a formal convergence proof of the Kalman filtering process.


Weakness:
(1) The method is sensitive to hyperparameters such as filter strength and step selection, particularly in tasks like object removal, where improper settings may introduce artifacts.

(2) The accelerated variant, Kalman-Edit*, relies on the assumption of linear behavior in latent space, which may not hold in more complex or nonlinear editing scenarios.

---

> ### Author Rebuttal · Authors · 2025-07-27
>
> Thank you for your valuable feedback. Below, we provide our detailed responses to the concerns you raised.
>
> [W1/Q] The sensitivity of our method to parameters
> - The term "sensitive" in our method refers to the introduction of additional Kalman control parameters compared to previous controller-based approaches. In practice, as long as a few key parameters, such as the added timestep and filter strength, are properly set, the overall editing performance remains strong.
> - To better show the sensitivity of our method to parameters, we conduct additional experiments. As shown in the table below, the previous controller-based method, RF-Inversion, even shows more sensitivity to strength and added steps. As these two parameters vary, the values produced by RF-Inversion show greater fluctuations compared to those generated by our approach. This suggests that the sensitivity of our method to parameter changes is lower than that of RF-Inversion.
>
> **CLIP-I Score Ablation Comparison on Zone Dataset**
> | Method                 | Filter or Controller Strength / Added Steps | 15-18  | 15-22  | 15-27      |
> | ---------------------- | ----------------------------- | ------ | ------ | ---------- |
> | **Kalman-Edit**  | **0.1**                       | 0.8770 | 0.9219 | **0.9346** |
> | **Kalman-Edit**  | **0.2**                       | 0.9043 | 0.9282 | 0.9226     |
> | **Kalman-Edit**  | **0.3**                       | 0.9014 | 0.8921 | 0.9079     |
> | **RF-Inversion** | **0.1**                       | 0.8613 | 0.9221 | **0.9311** |
> | **RF-Inversion** | **0.2**                       | 0.8921 | 0.9191 | 0.9259     |
> | **RF-Inversion** | **0.3**                       | 0.9174 | 0.8813 | 0.9063     |
>
> [W2] Difficulty of Kalman-Edit* handling complex editing scenarios
> - Kalman-Edit* indeed may not become practical in some editing scenarios. But it turns out to be efficient in most editing tasks, such as style transfer and scene editing (Please see the Appendix for more visualization results).

---

> > ### Comment · Reviewer_7uLu · 2025-08-05
> >
> > Thank authors for the extra results. I'll maintain my recommendation for acceptance.

---

### Note · Authors · 2025-08-13

We sincerely thank all reviewers for their valuable feedback. We are glad to learn that the reviewers recognize that our paper is well-motivated and has a good perspective (Reviewers 7uLu and bsYB), is grounded in control theory (Reviewers 7uLu, YwVg and ndBN), and shows good qualitative results (Reviewer YwVg). During the rebuttal process, we address their main concerns as follows:

* Hyperparameter sensitivity (Reviewer 7uLu): We show additional quantitative results where our method is less sensitive to hyperparameters compared to the baseline.
* Reason for better prompt-following score (Reviewer YwVg): We explain that the Zone and DIV2K datasets focus on local modifications. The design of our method allows for smaller estimation error, as evidenced by the higher local area similarity on the dataset.
* Insufficient qualitative results (Reviewer bsYB): To clarify that our method is not limited to local edits, we show an additional comparison of image stylization, in addition to the existing results on the HQ dataset containing large modifications (Figures 5 and 6). To address the concern about prompt following, we provide full edit prompts and will include more visual comparisons (in addition to Figures 2, 3 and 7 in the paper).
* Lack of advanced diffusion-based baselines (Reviewer ndBN): We provide additional quantitative comparisons with more advanced diffusion-based methods, demonstrating the advantages of our method.

---

### Decision · Program_Chairs · 2025-09-17

**Decision:**

Accept (poster)

**Comment:**

This paper proposes Kalman-Edit, framing rectified flow editing as a control problem and applying Kalman filtering to improve structural consistency. The connection to control theory in image editing is novel and well-motivated. Results across several datasets, with added ablations and baselines in rebuttal, show the method is effective.

The main weakness is qualitative: edits are often small, figures are low resolution, and comparisons are limited. While the rebuttal helped, the paper would benefit from clearer visual results and stronger baselines.

Overall, the contribution is solid and the method promising. The AC agrees with the majority of reviewers and recommends acceptance. Authors are encouraged to improve the qualitative results, provide higher-resolution figures, and include broader baseline comparisons in the camera-ready version.